



# Multipoint high-fidelity CFD-based aerodynamic shape optimization of a 10 MW wind turbine

Mads H. Aa. Madsen[1], Frederik Zahle[1], Niels N. Sørensen[1], and Joaquim R. R. A. Martins[2]

[1]Aerodynamic Design Section, DTU Wind Energy, Risø Campus, Frederiksborgvej 399, 4000 Roskilde, Denmark
[2]Department of Aerospace Engineering, University of Michigan, Ann Arbor, MI 48109, USA

**Correspondence:** Mads H. Aa. Madsen (mham@dtu.dk)

**Abstract.** The wind energy industry relies heavily on CFD to analyze new turbine designs. To utilize CFD further upstream the design process where lower fidelity methods such as BEM are more common, requires the development of new tools. Tools that utilize numerical optimization are particularly valuable because they reduce the reliance on design by trial and error. We present the first comprehensive 3D CFD adjoint-based shape optimization of a modern 10 MW offshore wind turbine. The optimization problem is aligned with a case study from IEA Wind Task 37, making it possible to compare our findings with the BEM results from this case study, allowing us to determine the value of design optimization based on high-fidelity models. The comparison shows, that the overall design trends suggested by the two models do agree, and that it is particularly valuable to consult the high-fidelity model in areas such as root and tip where BEM is inaccurate. In addition, we compare two different CFD solvers to quantify the effect of modeling compressibility and to estimate the accuracy of the chosen grid resolution and order of convergence of the solver. Meshes up to $14 \cdot 10^6$ cells are used in the optimization whereby flow details are resolved. The present work shows that it is now possible to successfully optimize modern wind turbines aerodynamically under normal operating conditions using RANS models. The key benefit of a 3D RANS approach is that it is possible to optimize the blade planform and cross-sectional shape simultaneously, thus tailoring the shape to the actual 3D flow over the rotor, which is particularly important near the root and tip of the blade. This work does not address evaluation of extreme loads used for structural sizing, where BEM-based methods have proven very accurate, and therefore will likely remain the method of choice.

## 1 Introduction

Wind turbine rotor optimization aims to maximize wind energy extraction and has been an important area of research for decades. A common metric is to minimize the levelized cost of energy (LCoE) (Ning et al., 2014), which decreases by lowering installation costs and operating expenses or by increasing the annual energy production (AEP). Simply upscaling the turbine leads to an increase in swept area, which in turn extracts more energy. However, a naïve upscaling does not capture the complexity of the problem (Ashuri, 2012).





A major drawback of naïve upscaling is that mass increases with the cube of the rotor radius. The industry avoids the prohibitive mass increase by improving the blade design, which has resulted in more slender blades for a given power rating, where the increase in loads (and therefore mass) can be kept low. This further results in blades with increased capacity factors.

Traditionally, the optimization process has been sequential, where the optimization of airfoils and subsequently planform are two distinct steps. In the present work we design the airfoils and the planform concurrently using 3D computational fluid dynamics (CFD). This concurrent design process is vital for the industry because as previously shown, concurrent design processes result in a larger gain compared to sequential counterparts (Barrett and Ning, 2018). This is the central principle in multidisciplinary design optimization (MDO) (Martins and Lambe, 2013).

The use of 3D CFD is particularly valuable near the turbine blade root and tip, since the blade element momentum (BEM) method uses empirical models to capture 3D effects for these regions. The increase in fidelity also allows us to explore out-of-plane features such as blade pre-bend and winglets, which is outside the scope of traditional BEM approaches.

Industry still relies heavily on BEM and for good reasons, given that 3D CFD shape design of rotors poses several challenges. One of these challenges is to model all the load cases that drive the design during an optimization. Much work has been done in steady state computations with steady, uniform inflow, but to truly generate realistic loads one should transition to turbulent inflow and accurately resolve the time domain. This poses an immense challenge in terms of memory and computation time and is an active area of research.

In this paper, we present results from a high-fidelity aerodynamic shape optimization of a 10 MW offshore wind turbine rotor. By "high-fidelity" we mean a detailed modeling of the rotor in 3D and the use of Reynolds-averaged Navier–Stokes (RANS) equations to model the aerodynamics throughout the optimization. The optimization is aligned with a case study from the International Energy Agency (IEA) Wind Task 37 [1], allowing for a comparison with the low-fidelity BEM results from this case study. Low-fidelity tools offer a fast and reliable modeling approach. However, BEM does not capture the physics as completely as high-fidelity CFD-based tools that solve the RANS equations. In the present work, we aim to quantify pros and cons for each approach.

Ideally, one would include all the interacting disciplines in such an optimization. This has been addressed by several previous works using BEM-based aeroleastic tools combined with various cross-sectional analytical or finite element based structural tools. Zahle et al. showed that simultaneous design of the aerodynamic shape and structural layout of a blade leads to passive load alleviation through torsional couplings Zahle et al. (2016) allowing significant increases in AEP without increasing loads and blade mass. LCoE has been minimized by other researchers while taking aerodynamics, structures, and controls into account, thereby truly treating it as an MDO problem both for 5 MW turbines (Ashuri et al., 2014) and for 20 MW turbines (Ashuri et al., 2016). Therefore, we aim to implement a high-fidelity structural solver as a natural next step, but for the present work we focus on aerodynamics alone.

---

[1]https://community.ieawind.org/tasks/taskdirectory





## 1.1 Related literature

CFD-based aerodynamic shape optimization is still rarely used in wind energy research, but both the aerospace and the automotive community have been using it increasingly. However, when it comes to low-fidelity shape optimization, the wind
energy community has a large body of work, some of which is briefly outlined below.

    BEM codes have been used extensively throughout the wind energy community for aerodynamic optimization. These codes are easy to implement and incur low computational costs. Robustness issues have been reported over time (Maniaci, 2011), which can be detrimental, especially when the analysis code is part of an optimization framework. This might not be as critical for researchers using gradient-free methods (Méndez and Greiner, 2006), but robustness is a key ingredient when gradients are
needed. To increase robustness, Ning (2014) proposed re-parameterization of the BEM-equations using a single local inflow angle, resulting in guaranteed convergence.

    It has long been known that the design of wind turbines is an inherently multidisciplinary endeavor. There have been more than two decades old research where BEM has been coupled with elastic beam models to account for structural analysis (Fuglsang and Madsen, 1999), including work in wind turbine optimization considering site specific winds (Fuglsang
and Thomsen, 1998, 2001; Fuglsang et al., 2002; Kenway and Martins, 2008). However, the efficient computation of gradients through the adjoint method to enable optimization with respect to large numbers of variables is a more recent development (Ning and Petch, 2016).

    BEM has also been coupled to structural models with different levels of fidelity. This allowed Bottasso et al. (2013) to study possible configurations to achieve bend-twist coupling resulting in load alleviation. They found that the highest load reduction
is obtained by combining (passive) bend-twist coupling and (active) individual pitch control instead of using only a single approach. Another example where BEM is part of a larger multidisciplinary tool applied to the study of load alleviation is that by Zahle et al. (2016). They maximized AEP without exceeding the original overall loads of a 10 MW reference wind turbine (RWT). They achieved a 8.7 % AEP increase through passive load alleviation without an increase in the blade mass and only minor increases in the loads, despite blades that were 9% longer. The parameterization comprises 60 design variables and just
as in the work by Bottasso et al. (2013), they computed the gradients with finite differences. After an initial step size study, they ran a reduced set of design load cases to obtain the final turbine design, which was then evaluated on the full design load basis. Their work is a demonstration of the power given by integrated design approaches.

    One obstacle in using BEM codes is that the lift and drag data must be at hand. Typically, one uses wind tunnel data or other low-fidelity methods, such as a panel code Kenway and Martins (2008), for this. Another option is to use high-fidelity
methods such as RANS CFD to generate the lift and drag coefficients for the BEM code. Barrett and Ning (2018) combine BEM with both panel and 2D RANS CFD in a comparison between two integrated blade design approaches ("precomputational" and "free-form") and a sequential approach. They used a panel code iteratively to converge the BEM residual and then subsequently either a panel code or CFD to generate the final lift and drag coefficients. Like Zahle et al. (2016), they argued for the integrated design approach but interestingly, they found that the precomputational approach achieved most of the benefits yielded by the



free-form approach. This is impressive since the precomputational approach only took marginally more computation time than the sequential approach.

Using a panel code on its own is also still an active area of research. An example of a panel-based optimization is the Risø-B1 airfoil family, which currently is in commercial use by several manufacturers. Fuglsang et al. (2004) described the

design and experimental verification process, where they used an in-house MDO tool. They carried out the numerical design studies using XFOIL (Drela, 1989) and used the VELUX wind tunnel for (2D) experimental verification. Due to concerns on XFOIL's accuracy with respect to separation, they opted to verify the optimization results using the CFD code EllipSys2D, thus combining fidelities in an attempt to balance speed and model accuracy.

Both gradient-based, gradient-free, and hybrid approaches have been used to optimize airfoils for the tip (Grasso, 2011) and

for the root (Grasso, 2012). More recent airfoil studies have turned to large offshore pitch-controlled wind turbines, including tests with vortex generators, that resulted in the development of a new airfoil family (Grasso, 2016).

## 1.2 High-fidelity CFD-based shape optimization

Barrett and Ning (2016) compared two numerical methods of different fidelities (a panel code and RANS CFD) to wind tunnel data. They found that the choice of aerodynamic analysis method had a large impact on the optimal design, which thereby

stresses the need for high-fidelity models. This agrees with Lyu et al. (2014), who report serious issues with Euler-based aircraft wing design due to missing viscous effects (compared to RANS-based design). They found that while Euler-based design yields some insights, the RANS-based optimization is needed to achieve a realistic design. Therefore, we limit the discussion in the present section to RANS CFD optimization.

### 1.2.1 Airfoil optimization

Kwon et al. (2012) used 2D RANS with a transition model to carry out gradient-based optimization using finite differences with 9 design variables and achieved a $11\%$ increase in torque. In the same vein, Ribeiro et al. (2012) used 9 design variables and a gradient-free method (genetic algorithm) to carry out both multi-objective optimization (50 generations) for 28 individuals and single-objective optimization (19 generations). By training a surrogate model, they achieved similar results while still gaining a speed-up of almost 50 %. Liang and Li (2018) used 2 design variables (thickness and camber) to carry out 2D shape

optimization with a gradient-free method of airfoils (NACA0015) for vertical axis wind turbines (VAWTs). A subsequent 3D modeling and CFD evaluation of the VAWT using the optimized airfoils achieved a power coefficient increases of up to 7 %. Finally, Zahle et al. (2014) carried out an airfoil optimization and wind tunnel validation. They used a combination of panel (XFOIL) and CFD (EllipSys2D) code for the analysis where the turbulence is described using the $k - \omega$ shear-stress transport (SST) turbulence model (Menter, 1993) and two transition models—$\gamma - \widetilde{Re_\theta}$ by Menter et al. (Menter et al., 2004; Langtry

et al., 2004; Sørensen, 2009) and the $e^N$ Drela–Giles transition model (Drela and Giles, 1986) described in (aag, 2002, Chapter 6). They used a total of 21 design variables and computed the gradient using finite differences. They ran 20 optimizations under various conditions and since each optimization comprised 2640 CFD simulations, they split the procedure in two steps of increasing fidelity to save time: First they optimized using XFOIL then they used this intermediate result as a starting



point for a subsequent CFD-based optimization. Such "hotstarts" are now common practice and we also leverage hotstarts in the present work. Using this framework, Zahle et al. (2014) completed the optimization of a 30 % and a 36 % airfoil called
LRP2-30 [2] and LRP2-36, respectively. Finally, through experimental results from the Stuttgart Laminar Wind Tunnel for both LRP2-30 and LRP2-360, as well as the FFA-W3 counterparts (FFA-W3-301 and FFA-W3-360), they demonstrated that the new airfoils exhibit a superior performance compared to the FFA-W3 airfoils.

### 1.2.2 Blade optimization

Shape optimization has also been carried out in 3D with CFD for gradient-free and gradient-based methods. Vucina et al.
(2016) used 3D RANS to shape optimize wind turbine blades with up to 25 design variables and a genetic algorithm. They concluded that their gradient-free setup has proven its functionality and robustness, but also that many generations were needed for the optimizer to converge due to the high number of variables.

As a final example of gradient-free methods in 3D CFD, Elfarra et al. (2014) optimized a winglet, also using a genetic algorithm. They used two design variables (cant and twist angle) to optimize the torque, resulting in a 9 % increase in power
production. The results were obtained by training a surrogate model (ANN) using 24 CFD samples to reduce computational time. There is an increasing interest in winglets, since a complete blade re-design can be too expensive. Winglets can be a less intrusive approach to retrofit the blade. Another CFD-based 3D winglet study is that by Zahle et al. (2018). They used 12 design variables to maximize the energy production while satisfying certain load constraints from the original blade. Like Elfarra et al., they also make used of a surrogate that they trained using a random sampling strategy. Here, the design is sought
to be more balanced by using multiple wind speeds throughout the sampling. Using gradient-based optimization, they obtained power increases of up to 2.6 % by extending the blade with a winglet and 0.76 % for straight blade extensions of 0.5% of the span. These results were achieved without increasing the flapwise bending moment at 90 % radius. In future work, they hope to include aeroelastic tailoring into the tip design for further improvements (Zahle et al., 2016). They find that the winglet is well suited to gain a power increase while limiting load increase. Interestingly, the winglet itself does not directly contribute
to the power production but it diffuses the tip vortex, which in turn reduces the induced drag further inboard where the power increase is observed.

In the above work, when gradients were computed, they were approximated by finite differences (Fuglsang et al., 2002; Bottasso et al., 2013; Zahle et al., 2016; Fuglsang et al., 2004; Kwon et al., 2012; Zahle et al., 2014, 2018). The cost of finite-difference gradients scales linearly with the number of design variables so there is a need for more efficient methods when
handling design problems with large dimensionality. The complex-step derivative approximation method is an alternative to finite differences that is much more accurate, but still scales linearly with the number of variables (Martins et al., 2003). This method has been widely used, including some wind energy applications (Barrett and Ning, 2018; Kenway and Martins, 2008). Some efforts tried to reduce the computational cost by using semi-empirical gradients (Fuglsang and Madsen, 1999), surrogate models (Ribeiro et al., 2012; Elfarra et al., 2014; Zahle et al., 2018), and mixed fidelity models (Barrett and Ning, 2018; Zahle et al., 2014). Recent advances using symbolic differentiation, automatic differentiation (AD) and the adjoint method (Barrett

---

[2]LRP stands for Light Rotor Project





and Ning, 2018; Ning and Petch, 2016; Barrett and Ning, 2016) result in both time saving *and* accuracy improvement, but the required implementation effort is significant. The accuracy of the gradients for the adjoint method inherently depends on the convergence of the flow field. Many of the works cited below work with steady state RANS equations and face convergence issues due to unsteady flow phenomena, so this issue must be handled in the adjoint implementation.

The number of design variables can indeed become troublesome for the predominant methods listed above (gradient-free and finite differencing). In the benchmark by Lyu et al. (2014) mentioned earlier, they test gradient-free and gradient-based methods on the multidimensional Rosenbrock function, a RANS-based twist optimization and an aerodynamic shape optimization problem and conclude that "gradient-based algorithms are the only viable option for solving large-scale aerodynamic design optimization problems". Indeed, they find that the gradient-free methods require two to three orders of magnitude more computational effort. The above cited works use a number of design variables well below 100 and still struggle to reduce computational costs. This is in stark contrast to the efforts using the adjoint method cited below, where up to several hundred design variables are commonly optimized. Interestingly, the findings by Lyu et al. (2014) seem to resonate with more recent work from the wind energy community (Vorspel et al., 2017), and both works point to the use of the adjoint method for an efficient gradient computation.

### 1.2.3 High-fidelity optimization using the adjoint method

In the following we only highlight works that contain RANS CFD-based shape optimization using the adjoint method. Note that we already cited works including 2D RANS CFD optimization using the adjoint method (Barrett and Ning, 2018, 2016) via the open-source, compressible solver SU2 (Palacios et al., 2013), which we will not redundantly cite again. It should also be noted that this section is not intended to be exhaustive but merely serves to give the reader a familiarity with known works within the wind energy field.

Ritlop and Nadarajah (2009) were the first to use a high-fidelity shape optimization method with an adjoint solver for wind turbine profiles. They optimize the lift to drag ratio for S809 airfoils using a compressible solver, a low-Mach preconditioner (both for flow and adjoint solver) and the Spalart–Allmaras (SA) turbulence model and find a tendency to increase camber to gain more lift. Finally, they point to the $k-\omega$ SST turbulence model and a transition model as needed improvements. A related work by Khayatzadeh and Nadarajah (2011) later presents optimization results on the same airfoil using an enhanced framework including said improvements. Here, they attempt to postpone onset of transition. They conclude that the capability and accuracy of the discrete adjoint optimization framework is improved.

After Khayatzadeh and Nadarajah's work there have been several interesting contributions to 2D RANS shape optimization by Schramm et al. (Schramm et al., 2014, 2015, 2016, 2018) who use the continuous approach. In these works the continuous adjoint implemented for ducted flows in the flow solver OpenFOAM (Weller et al., 1998) is extended to external aerodynamics. First (Schramm et al., 2014), they optimize lift to drag ratio of the DU 91-W2-250 profile using 720 design variables. Here, they use the 'frozen turbulence' assumption, which means that no adjoint equation is derived (and solved) for the turbulence model. Since each surface point in this work is a design variable they smooth the gradient for stability. The result is a $5.7\%$, $24.0\%$ and $59.0\%$ increase in lift over drag ratio for AoAs: $6.15°, 8.18°$ and $9.66°$. Finally, they demonstrate their setup can incorporate





geometric cross-section area constraints. In a later work, Schramm et al (Schramm et al., 2015, 2016) present finite-difference verification of the adjoint gradients. One can also find shape optimization of an upstream leading edge slat (Schramm et al., 2016) for the DU 91-W2-250 airfoil and a validation of the computational setup using wind tunnel data. The results show good

agreement below stall. The optimization results in a 2 % decrease in drag for $\mathrm{Re} = 6 \cdot 10^5$ and is obtained with 480 design variables.

In a more recent work by Schramm et al. (2018) they investigate the effect of the "frozen turbulence" assumption in 2D. They carry out their investigations on NACA-0012 and DU 93-W-210 airfoils. It is an unconstrained, single point study and they conclude that the implementation of adjoint turbulence models results in better gradients than those obtained through the

frozen turbulence assumption. Finally, they specifically mention thickness constraints as a future work topic.

The OpenFOAM setup with a continuous adjoint solver has also been used in 3D. This was done by Vorspel et al. (2016) who first carry out a 2D test case with two design variables, which they find to be fast with a stable convergence. The 3D test case is an extruded airfoil with the spanwise length equal to 5 chords and a mesh of $2.4 \cdot 10^6$ cells and a $y^+$ of 2.5. They investigate both a twist and a bend-twist-coupling case but find that the bending has no visible effect. Something they expect to

change for future rotating blades applications. The briefly cited benchmark paper by Vorspel et al. (2017) is indeed also with OpenFOAM but it is kept in 2D.

Besides the above mentioned work by Vorspel et al. (2016), which does not include rotation there are (at least) four important works within this research field in 3D, namely the work by Economon et al. (2013), Dhert et al. (2017), Vorspel et al. (2018), and Tsiakas et al. (2018). The first three all studied the NREL Phase VI rotor whereas Tsiakas et al. (2018) studied the MEXICO

rotor. The earliest work of the three is that by Economon et al. who use a continuous adjoint formulation to perform single point aerodynamic shape optimization using a compressible RANS model. In 2D they reduce drag on a NACA 4412 profile by $4.86\%$ under imposed thickness constraints to account for the structural aspect. They use a total of 50 design variables and complete 10 design iterations to obtain the drag reduction. In 3D, they improve the torque coefficient on a mesh comprising $7.9 \cdot 10^6$ cells with $4\%$ using 84 shape variables for each Free-Form Deformation (FFD) box side. We note that only complete

3 design iterations. One drawback in this early work is the use of the frozen turbulence assumption, which they also identified as an area of future work.

Dhert et al. (2017) use a discrete adjoint solver to carry out a multipoint optimization on a $2.6 \cdot 10^6$ cell mesh where they optimize the torque coefficient using up to 252 design variables. They use pitch, twist and local shape design variables while they constrain the thickness between 15 % and 50 % of the local blade chord to ensure adequate place for a structural box. The

final multipoint optimization results in a 22.1 % increase in torque coefficient. They find the optimized shape both for single and multipoint optimization to exhibit highly cambered trailing edges at the root region where the wind speed is reduced. While this does agree with what has been reported in 2D cases (Ritlop and Nadarajah, 2009) it is also exactly what one would expect when chord is not included as a design variable. It should be mentioned that the work by Ritlop and Nadarajah (2009) is relevant since they studied the S809 airfoil on which the entire blade span is based. Finally we note that the original design was meant as a three bladed rotor, which explains the low thrust coefficients in the reported results (Dhert et al., 2017, Tab. 1).



Since the present work can be seen as a natural next step from the work by Dhert et al. we briefly outline some of the improvements made, which are also summarized in Tab. 1: First of all, we implemented the use of reverse AD to assemble the

adjoint equation when handling rotational setups to avoid the time costly forward AD assembly. We verified the reverse AD gradients with finite difference and checked the consistency between forward and reverse AD gradients. We also added constraints on maximum thrust and flapwise bending-moment to align with the IEA case study and enlarged the design space to include e.g. chord as a design variable whereas Dhert (Dhert, 2015, p. 77) ruled it out due to mesh failure considerations. Furthermore, Dhert et al. carried out their studies on a reduced geometry because of flow solution convergence issues whereas

we now can include the entire geometry. Finally, we mention the improved convergence of the design optimization problem where we now typically solve cases down to below $10 \cdot 10^{-5}$ whereas convergence for Dhert et al. seems to lie somewhat higher (Dhert et al., 2017, tab II), namely around; $[10 \cdot 10^{-1}, 10 \cdot 10^{-2}]$.

The improved convergence can be seen as a testimony to the work done by MDOLab [3] to continuously upgrade their tool chain. Here, especially the change in CFD solver should be noted: The solver used by Dhert et al. called SUmb (Weide et al.,

2006) has been further developed into the CFD solver now known as ADflow, featuring numerous improvements including the recently implemented Approximate Newton Krylov (ANK) solver, which is used in the present work. The various components in the optimization framework will be described in section 2.

**Table 1.** Overview of differences between the work by Dhert et al. (2017) and the present work.

|  | **Dhert et al.** | **Present work** |
| --- | --- | --- |
| Geometry | Reduced geometry (no root) | Entire geometry included |
| Adjoint solver | Forward AD with coloring | Reverse AD |
| Convergence of design problem | $10^{-1}$ | $10^{-4}$ |
| Design iterations | $\mathcal{O}(10^1)$ | $\mathcal{O}(10^2)$ |
| Mesh resolution † | $< 2.60 \cdot 10^6$ | $< 14.16 \cdot 10^6$ |

† Largest mesh used for optimization

The second most recent work we mention is that by Vorspel et al. (2018) who present unconstrained optimization of the

NREL VI rotor where they minimize the thrust by varying up to 9 twist design variables using the steepest descent algorithm. Like other mentioned approaches they use a projection method to reduce the design space from number of surface mesh points

---

[3]http://mdolab.engin.umich.edu/





to number of design variables. They compare the obtained gradients to the spanwise thrust and reassuringly they find similar trends in magnitude of the two distributions as one would expect from a functioning setup. Not surprisingly, they mention convergence issues among other things owed to the turbine being stall regulated exhibiting separated flow at an inflow speed of $10\ m/s$. Vortices at tip and root further impair the convergence, which in turn results in poor gradient quality but by limiting

the deformable area to only 50 % of the blade length (from $[2.00\ m, 4.75\ m]$ out of a 5 m span) they limit the vortex influence. We note that this rules at any deformation (i.e. twisting) of the tip. Like Economon et al. they include the entire geometry and use the frozen turbulence assumption. They do however differ in choice of turbulence model where Vorspel et al. use the $k-\omega$ SST model. For future work they point to the use of more efficient optimization algorithms. They also mention the inclusion of adjoint turbulence equations and the study of turbines, which are not stall regulated.

The final and most recent work we mention within shape optimization applied to wind energy applications is that by Tsiakas et al. (2018). They use a continuous adjoint approach including turbulence equations derived from the SA model to optimize the MEXICO RWT. The flow is modelled by the incompressible RANS equations where the flow is solved in a co-moving frame of reference and their chosen objective function is to maximize power for a single wind speed of $10\ m/s$. Compared to the present work they use a different parameterization technique based on volumetric Non-Uniform Rational B-Splines

(NURBS). Noticeably, the NURBS confine the blade in a small volume and they are used both for the deformation of surface and volume mesh. The outermost NURBS control points are locked to keep the outer volume mesh fixed. We note that this only ensures $C^0$ continuity. They use 385 NURBS resulting in 135 design variables which are only allowed to move in the streamwise direction. We note, that this choice of parameterization limits the design space and e.g. no chord increase can be obtained without simultiously changing profile shape. The remarkable thing in this work is that flow and adjoint analysis is

sucessfully ported on Graphics Processing Units (GPUs) resulting in sizeable speed-up. They obtain a $3\ \%$ increase in objective function and ascribe the minor improvement to the limited freedom in the parameterization.

Summing up, we have presented a few pioneering works, which first in 2D then in 3D successfully apply the adjoint method for CFD shape optimization within wind energy albeit many improvements are needed before the industry can be provided with a 'push-button-solution'. There is (to the authors' knowledge) no published works on high-fidelity 3D RANS-based shape

optimization where the whole rotating geometry of a rotor is modeled while important areas of improvements such as

i) inclusion of relevant structural constraints,

ii) a deep, stable convergence of flow and adjoint variables

iii) inclusion of turbulence models and

iv) a comprehensive set of design variables

have been simultaneously addressed. We hope the present work can serve a purpose to this end. Below, we have summarized the above discussed literature in Table 2 for the convenience of the reader.

Finally, at the end of this short literature overview we allow a side note to briefly mention the recent work by Anderson et al. (2018) who couple the high-fidelity NSU3D flow solver and the high-fidelity AStrO structural finite element solver through



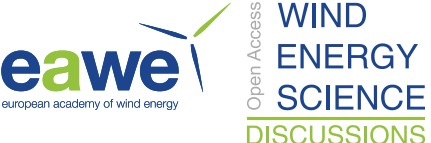

**Table 2.** Overview of related works using the adjoint method. In the above literature review we also mention a benchmark study by Vorspel et al. (2017) but since they use a variety of algorithms against we refrain from adding them below and instead refer to the paper for further details.

| Author | Year | Adjoint approach | Dimension | Mesh size † | Design variables | Design cycles § |
|---|---|---|---|---|---|---|
| Ritlop and Nadarajah (2009) | 2009 | Discrete | 2D | $0.32 \cdot 10^5$ | 385 †† | 100–200 |
| Khayatzadeh and Nadarajah (2011) | 2011 | Discrete | 2D | $1.31 \cdot 10^5$ | 385 | - |
| Schramm et al. (2014) | 2014 | Continuous | 2D | $55.00 \cdot 10^3$ | 720 | - |
| Schramm et al. (2016) | 2016 | Continuous | 2D | - | 480 | - |
| Barrett and Ning (2016) | 2016 | Continuous ¶ | 2D | $14.00 \cdot 10^3$ | 10–22 [Tab. 2] | - |
| Schramm et al. (2018) | 2018 | Continuous | 2D | $2.08 \cdot 10^5$ | 20-50 | 0–30 [Fig. 5,7] |
| Barrett and Ning (2018) | 2018 | Continuous ¶ | 2D | $14.34 \cdot 10^3$ | 10–68 [Tab. 1] | - |
| Economon et al. (2013) | 2013 | Continuous | 2D | $32.0 \cdot 10^3$ | 50 | 10 |
| | | | 3D | $7.90 \cdot 10^6$ | 84 | 3 |
| Vorspel et al. (2016) | 2016 | Continuous | 2D | - | 2 | 30 [Fig. 3] |
| | | | 3D | $2.4 \cdot 10^6$ | - | <8 [Fig. 6] |
| Dhert et al. (2017) | 2017 | Discrete | 3D | $2.60 \cdot 10^6$ ‡ | 1-252 | 9-23 |
| Vorspel et al. (2018) | 2018 | Continuous | 3D | $5.20 \cdot 10^6$ ‡‡ | 5-9 | <8 [Fig. 5] |
| Tsiakas et al. (2018) | 2018 | Continuous | 3D | $2.50 \cdot 10^6$ §§ | 135 | 10 [Fig. 4] |
| Madsen et al. (present work) | 2018 | Discrete | 3D | $14.16 \cdot 10^6$ | 1-154 | 100-200 |

† Number of cells in largest mesh used for optimization

†† The number of design variables is not explicitly stated but gleaning to the PhD thesis from Nadarajah (Nadarajah, 2003) there are evidently two parameterization options: i) mesh points (seemingly the preferred approach by Nadarajah) and ii) Hicks–Henne functions. Since the latter do not require gradient smoothing and since Ritlop and Nadarajah do indeed smooth gradients we assume that they use the actual surface mesh points as design variables. Given that they use a C-mesh and a later, related work for a C-mesh of similar chordwise resolution have 385 nodes (Khayatzadeh and Nadarajah, 2011) on the airfoil surface we find this number to be a reasonable estimate.

§ This column is the most depleted one given that not all papers clearly state the amount of steps taken. In some case we give number of design cycles read from figures, which are then cited. Furthermore, the number greatly depends on the defined optimization problem and optimizer settings meaning that cross-setup comparison is difficult. However, it is included for interested readers for the sake of completeness.

§§ Tsiakas et al. (2018) only give the number of mesh nodes.

‡ While both Dhert et al. (2017), Economon et al. (2013) and Vorspel et al. (2018) modeled the NREL VI turbine we note that the work by Dhert et al. was on a reduced geometry where the root section was removed.

‡‡ Notice their use of symmetric BCs effectively doubling the grid resolution when comparing to e.g. the work by Dhert et al.

¶ The specific adjoint solver architecture they use for SU2 is not stated but the paper they use as reference (Palacios et al., 2013) from 2013 specify that the continuous adjoint solver is implemented and that the discrete and hybrid adjoint solvers are under development.



a fluid structure interface to converge on realistic, steady-state loads on the SWiFT RWT. Subsequently, a purely structural optimization is carried out with the computed loads. Here, an impressive 16310 structural design variables (ply orientation) are used in a gradient-based optimization. They complete 10 optimization design cycles for 5 different load cases and observe a

40-60 % reduction in maximum fatigue stress criterion. As it is a structural optimization the work by Anderson et al. is slightly outside the scope of the present work where we focus on aerodynamic shape optimization. However, they do point to including the aerodynamics in a fully coupled aerostructural optimization as a natural next step, which is exactly the same next logical step as for the present work.

### 1.3   IEA Task 37 case study

The IEA Task 37 aims to help coordinate international research activities within wind energy and to this end it focuses on establishing reference systems (turbines or plants) and various benchmark activities within multidisciplinary analysis and optimization (MDAO). It can be useful to view MDAO through a 3D prism (Perez-Moreno et al., 2016, Fig. 5) where the

dimensions are i) model fidelity, ii) architecture and iii) system scope. Here, it is hoped that the IEA benchmarks span all the axis and since there only have been contributions from BEM codes for the benchmark in question, we can now with the present high-fidelity work explore the axis i) of (increased) model fidelity. In section 4 we highlight exactly how we have implemented the case study and which deviations we see between the problem definition for the BEM codes and the present work.

### 1.4   Overview of presented work

We start the remainder of this paper with a section on methodology (Sec. 2), followed by a comparative analysis in Sec.3 between the compressible flow solver and an in-house incompressible flow solver, since no experimental data for the turbine is available. After the analysis, the actual design optimization problem is presented in Sec. 4, leading to the presentation of the optimization results in Sec. 5 before a final conclusion of the case study is given in Sec. 6.

### 2   Methodology

We now briefly describe all components in the optimization framework. The overall workflow can be seen in Fig. 1 where we have borrowed visual setup from the extended structure matrices (Lambe and Martins, 2012). In short, an initial set of design variables, $\mathbf{x}^{(0)}$, is given to the optimizer, which then passes the variables on to the surface deformation module prompting it to update the surface mesh. This module also provides analytic derivatives of the surface mesh with respect to the design variables, $d\mathbf{x}_s/d\mathbf{x}$. After the surface mesh has been updated it is passed to the volume deformation module, which updates the volume mesh and also is capable of returning analytic derivatives $d\mathbf{x}_v/d\mathbf{x}_s$, which are the derivatives of the volume mesh with respect to the surface mesh. Now, the mesh is ready and the flow solver can be called to compute the flow states, $\mathbf{w}$. These are in turn passed on to the adjoint solver, which (using reverse AD) solves the adjoint equation with PETSc (Balay et al., 2018a,





b, 1997) and computes the total derivative. Finally, the function of interest $f$ (e.g. torque) as well as its derivative, $\mathrm{d}f/d\mathbf{x}$, can

5  be passed to the optimizer and a new step can be taken.

The shape optimizations presented below involve $\mathcal{O}(10^2)$ major iterations, which also is the minimum bound of numbers of consecutive CFD RANS solutions and mesh updates (there will be additional state solves due to line searches). The many iterations result in extreme pressure on every component in the tool chain not only on high performance but also on robustness such that the optimization does not crash mid way. Even for the mesh deformation steps, which perhaps are the easier components in Fig. 1, this means robust and differentiable steps on meshes with up to $\mathcal{O}(10^7)$ cells.

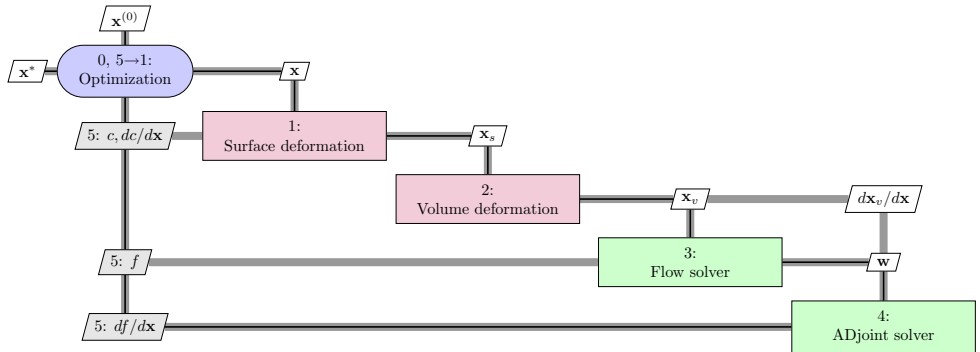

**Figure 1.** An extended design structure matrix (XDSM) showing the optimization framework. Green blocks are usually iterative analysis codes whereas red boxes describe functions. Both the surface and volume deformation steps are fast, explicit operations. On the other hand, the flow solver and the adjoint solver are time consuming, iterative operations the take up the vast majority of the computation time.

## 2.1 Mesh and Geometry manipulation

### 2.1.1 Surface deflection: pyGeo

To deform the surface mesh we use the Python module pyGeo developed by Kenway et al. (2010), which is based on the well established FFD (Sederberg and Parry, 1986) technique. Some of the key, salient features in FFD are analytic derivatives

15  and exact (to machine precision) representation of the initial geometry provided that an inverse Newton search has been well converged. We refer interested readers to the original work by Kenway et al. (2010) for further information.

### 2.1.2 Volume mesh deformation: IDWarp

The volume deformation tool is called IDWarp and is based on the work by Edward et al. (2012) who present a deformation method using the inverse distance weighting function. IDWarp (previously known as pyWarpUstruct) is a fast and unstructured deformation algorithm used in a large body of work from the MDOLab at the University of Michigan. We refer to Edward et al. (2012) for implementation details.





### 2.2 Flow solvers

#### 2.2.1 EllipSys3D

EllipSys3D is a in-house, structured, multiblock, Finite Volume Method (FVM) flow solver developed at DTU Wind Energy
by Michelsen (1992, 1994) and Sørensen (1995), which we use in the present work to carry out the comparative analysis. It
discretizes the incompressible RANS equations using general curvilinear coordinates and couples velocity and pressure through
the SIMPLE algorithm. The state variables are collocated (all stored at cell center). As is known, this can lead to odd-even
decoupling, which results in checkerboard patterns in computed fields. In EllipSys3D, this is countered using the Rhie/Chow

interpolation (Rhie and Chow, 1983). Even though ADflow only has a second order scheme we will for EllipSys3D use the
third order quadratic upwind interpolation for convection kinematics (QUICK) scheme. Finally, we use the $k-\omega$ SST (Menter,
1993) model to calculate the turbulent eddy viscosity, which compares favorably to other turbulence models for wind turbine
applications (Reggio et al., 2011).

EllipSys3D has been validated against experimental data both for the MEXICO RWT (Bechmann et al., 2011) as well as

for the oft mentioned NREL Phase VI RWT (Sørensen et al., 2002; Sørensen and Schreck, 2014) and participated in a blind
comparison (Simms et al., 2001). For the latter RWT also the unsteady interaction between tower and blade has been simulated
using overset grid capabilities where an overall good agreement was found with experimental data (Zahle et al.). The solver fur-
thermore figured in a large body of work for rotor applications concerning e.g. aerodynamic power (Johansen et al., 2009) and
fluid-structure-interaction (Heinz et al., 2016) to name but a few. Incidentally, the latter work also encompasses a comparison

across fidelities between the CFD-based tool, HAWC2CFD, and the BEM-based HAWC2 solvers where a good agreement was
found. Finally, we mention that EllipSys3D has been (favorably) compared to the open source solver, OpenFOAM, mentioned
in the literature overview above (Cavar et al., 2016).

#### 2.2.2 ADflow

ADflow is an in-house, compressible RANS solver developed at the MDOLab and is used in all presented optimizations.

ADflow is based on SUmb (Weide et al., 2006) and is a structured FVM CFD solver using cell centered variables on a
multiblock grid that is written in Fortran 90 and wrapped with Python to provide an intuitive userinterface. Early works on
SUmb's adjoint solver implemented for the Euler equations is presented by Mader et al. (2008). Said work was then extended
to take rotational physics into account as described by Mader and Martins (2011). A few years later, Lyu et al. (2013) finally
presented an improved adjoint solver (using forward AD) that could model the RANS equations for rotational physics.

Unlike EllipSys3D, ADflow uses the Spalart-Allmaras (SA) (Spalart and Allmaras, 1994) turbulence model and works with
state variables computed using the Jameson-Schmidt-Turkel (JST) scheme. ADflow has a built in adjoint solver to provide
gradients and is along with TACS the cornerstone in the MDO of aircraft configurations with high-fidelity (MACH) frame-
work (Kenway and Martins, 2014) used at the MDOLab for studies on entire aircraft configurations resulting in numerous

high-fidelity optimizations (Lyu et al., 2014; Dhert et al., 2017; Kenway and Martins, 2014). As mentioned, we use ADflow's
ANK solver. In short, ANK counters converge issues for standard Newton-Krylov (NK) methods due to starting points being





remote from the basin-of-attraction by using a globalization method called pseudo-transient-continuation, which through a backward Euler method inherits stability. Once removed from the starting point they raise the time step to approach the higher convergence rate associated with an NK solver.

The ANK method involves the solution of large linear systems using preconditioners. These are solved in a matrix-free operation with the GMRES (Saad and Schultz, 1986) algorithm using the PETSc library (Balay et al., 2018a, b, 1997). This is also true for the linear systems encountered in the adjoint solver. ADflow is considered converged at the n'th iteration if a ratio of the residual, $\mathcal{R}^n$, is below a given tolerance, $\eta_{abs}$, where the ratio is formed using the free stream residual; $\mathcal{R}^{fs}$.

$$\eta_{abs} \leq \frac{||\mathcal{R}^n||_2}{||\mathcal{R}^{fs}||_2} \tag{1}$$

For the optimizations presented below we typically set $\eta_{abs} = 10^{-12}$ whereas the L2-convergence for the adjoint equation is set one or two orders of magnitude above. These convergence thresholds are not to be confused with the threshold for the optimizer called `Major optimality tolerance`, which we set to $10 \cdot 10^{-5}$.

## 2.3    ADjoint: A discrete adjoint solver architecture

After we have found a converged flow solution with ADflow we can compute any function of interest, $f(\mathbf{x}, \mathbf{w})$, from the mesh,
$\mathbf{x}$, and the flow field, $\mathbf{w}$. To get the gradient, $df/d\mathbf{x}$, we start with the equation for the total derivative:

$$\frac{df}{d\mathbf{x}} = \frac{\partial f}{\partial \mathbf{x}} + \frac{\partial f}{\partial \mathbf{w}}\frac{d\mathbf{w}}{d\mathbf{x}}, \tag{2}$$

and use that not only the residual, $\mathbf{R}$, but also its derivative, $d\mathbf{R}/d\mathbf{x}$, should be zero:

$$\frac{d\mathbf{R}}{d\mathbf{x}} = \frac{\partial \mathbf{R}}{\partial \mathbf{x}} + \frac{\partial \mathbf{R}}{\partial \mathbf{w}}\frac{d\mathbf{w}}{d\mathbf{x}} = 0, \tag{3}$$

to substitute the solution Jacobian, $d\mathbf{w}/d\mathbf{x}$, into equation (2):

$$\frac{df}{d\mathbf{x}} = \frac{\partial f}{\partial \mathbf{x}} - \underbrace{\frac{\partial f}{\partial \mathbf{w}}\left[\frac{\partial \mathbf{R}}{\partial \mathbf{w}}\right]^{-1}}_{\Psi^T}\frac{\partial \mathbf{R}}{\partial \mathbf{x}}. \tag{4}$$

The linear system in equation (4) can either be solved by computing the solution Jacobian, $d\mathbf{w}/d\mathbf{x}$, from the tangent linear system (3) or by solving the adjoint system:

$$\left[\frac{d\mathbf{R}}{d\mathbf{w}}\right]^T \Psi = \left[\frac{df}{d\mathbf{w}}\right]^T \tag{5}$$

to get the adjoint variables, $\Psi$, which in turn can be used to find the total derivative:

$$\frac{df}{d\mathbf{x}} = \frac{\partial f}{\partial \mathbf{x}} - \Psi^T\frac{\partial \mathbf{R}}{\partial \mathbf{x}}. \tag{6}$$

Evidently, one should use the adjoint method when many design variables are at play, since $\mathbf{x}$ is not part of the adjoint equation (5) whereas the tangent (or forward) approach is preferable when dealing with many functions of interests. Given that we work with $\mathcal{O}(10^2)$ design parameters but only a few functions of interests we prefer the adjoint method. As mentioned in the
introduction, we have upgraded the framework for rotating setups to exclusively use reverse AD when assembling all four partial derivative matrices needed to solve equations (5) and (6).





## 2.4 Optimizer

We use the Sparse Nonlinear OPTimizer (SNOPT) (Gill et al., 2002) based on a sequential quadratic programming (SQP) algorithm in all optimizations. Like all other components in the framework it is wrapped in Python (through pyOptSparse) allowing for an intuitive user interface. As already noted, the convergence in SNOPT is set through the `Major optimality tolerance` variable. We aim at converging all optimization problems down to $10 \cdot 10^{-5}$ (or $1.0E-4$ in SNOPT terminology). Interested readers are referred to the SNOPT manual [4] for further details.

## 5  3  Analysis

To verify that ADflow is set up correctly we first perform a comparison with EllipSys3D. As the mesh is refined we expect the two solvers to converge towards the same, identical solution. The case study is defined with a cut-in speed of $4\,m/s$ and a cut-out speed of $25\,m/s$. Within this range we use the eight operational conditions defined in Tab. 3 to compare the solvers.

**Table 3.** Operational conditions for the simulations in the analysis. Notice that for the compressible solver, ADflow, we use velocity, density and temperature as input parameters. ADflow then computes the complete thermodynamic conditions with said input. Thus, e.g. pressure is computed using the relation: $p = \rho \cdot T \cdot R$ where $\rho$ is density, $T$ is temperature and $R$ is the gas constant for dry air: $R = 287 J/(K \cdot kg)$. Density is set to the density of air at sea level and 15 °C, $\rho = 1.225\,[kg\,m^{-3}]$ and dynamic viscosity is set to $\mu = 1.784 \cdot 10^{-5}\,[kg\,m^{-1}\,s^{-1}]$.

| Run | Wind speed | RPM | Rotation rate †, $\omega$ | Pitch |
|-----|------------|-----|---------------------------|-------|
|     | $[m\,s^{-1}]$ | $[-]$ | $[rad\,s^{-1}]$ | $[deg]$ |
| wsp04_L0 | 4.0 | 6.00 | 0.63 | 0 |
| wsp06_L0 | 6.0 | 6.00 | 0.63 | 0 |
| wsp08_L0 | 8.0 | 6.69 | 0.70 | 0 |
| wsp10_L0 | 10.0 | 8.36 | 0.88 | 0 |
| wsp11_L0 | 11.0 | 9.20 | 0.96 | 0 |
| wsp12_L0 | 12.0 | 9.60 | 1.01 | 0 |
| wsp15_L0 | 15.0 | 9.60 | 1.01 | $6.74 \cdot 10^{0}$ |
| wsp25_L0 | 25.0 | 9.60 | 1.01 | $1.90 \cdot 10^{1}$ |

† Based on targeting a tip speed ratio, $\gamma$, of 7.8 where $6.0 \leq RPM \leq 9.6$.

---

[4]http://www.sbsi-sol-optimize.com/manuals/SNOPT%20Manual.pdf



## 3.1 Computational mesh

All simulations are 3D CFD steady state, rotor-only computations where effects from both tower and nacelle have been neglected. Given that we study an upwind turbine this negligence should have a limited effect. We also note that we compute the flow field using a co-rotating, non-inertial reference frame, which is attached to the rotor. The RANS equations therefore have additional terms to account for Coriolis and centripetal forces. Just as for the IEA Task 37 the three-bladed, pitch regulated rotor geometry in the analysis is a perturbed design based on the DTU 10 MW RWT (Bak et al., 2013) where both chord and twist distributions have been altered to allow for more room for improvement in the ensuing optimization. Fig. 2 compares the DTU 10 MW RWT and the perturbed design.

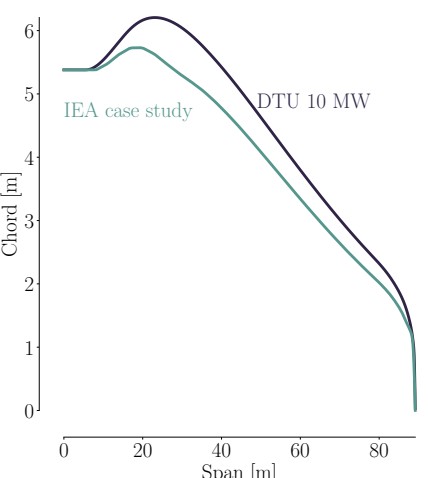 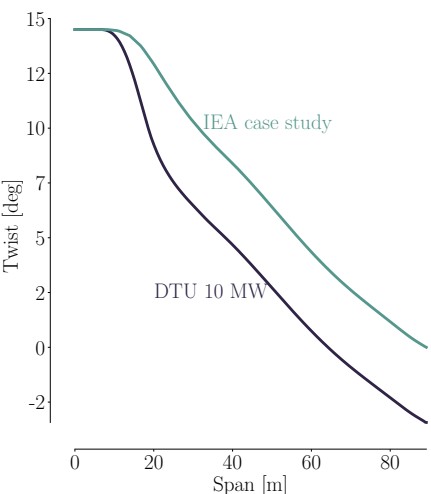

**Figure 2.** Comparison of chord (LHS) and twist (RHS) for the DTU 10 MW RWT and the perturbed design used as the starting point for optimizations in the IEA case study. As seen, the chord is somewhat reduced and the twist is less aggressive. The initial design is based on the FFA-W3 airfoil family with relative thicknesses in the range $[24\%, 36\%]$ where higher thicknesses are interpolated.

### 3.1.1 Surface and volume mesh

The surface mesh consists of three blades each with 36 blocks. For each blade there are 256 cells in the chordwise direction and 128 in the spanwise direction (tip excluded). The surface mesh was generated using the in-house Parametric Geometry Library (PGL). The tip was constructed using 4 blocks of $32 \times 32$ cells each. Indeed, all blocks are square $32 \times 32$ cell blocks resulting in a total surface comprising 110592 mesh cells. The spherical volume mesh is an O-O topology generated with the hyperbolic in-house mesh generator HypGrid (Sørensen, 1998). Setting the first boundary layer cell height to $1 \cdot 10^{-6}$ m yields a $y^+$ around 1 for the given operational conditions and in total 128 cell layers are grown from the surface mesh with the farthest vertices reaching a distance of $1740$ m. This results in a total of 432 blocks each of $32 \times 32 \times 32$ cells, which is equivalent to $14.155776 \cdot 10^6$ cells. Given a span of $R = 89.166$ m the surrounding spherical mesh expands about 20 times the blade span.

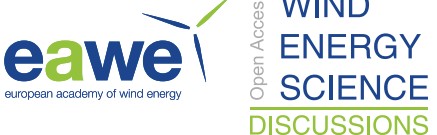



The mesh described above is called L0. The L1 mesh is obtained by coarsening L0 once, i.e., by removing every second cell in all three directions. L2 is obtained by coarsening L1 and so on. Unless otherwise stated we use these three meshes in all presented work. A visualization of the turbine geometry and the surrounding spherical mesh can be seen in Fig. 3 and a zoom of the rotor is given in Fig. 4.

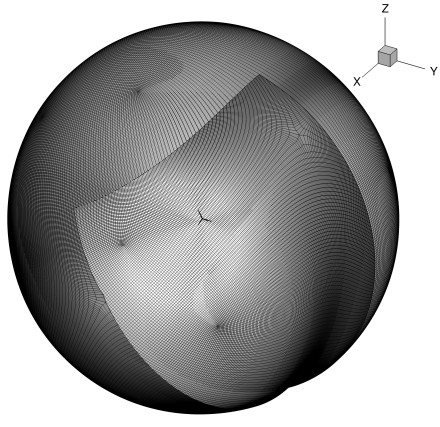

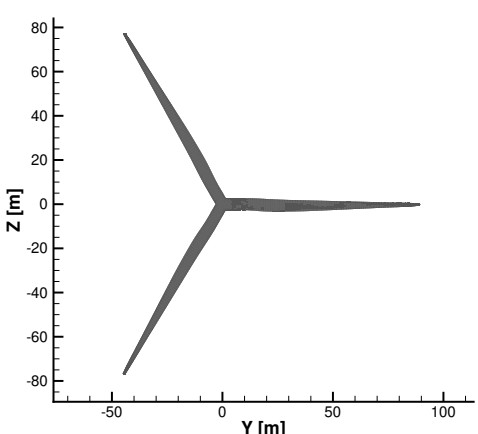

**Figure 3.** The initial wind turbine design at origin with the spherical L0 mesh around it. The blade span is $89.166\ m$ and the spherical mesh stretches to $1740\ m$ giving a ratio close to 20 times the blade span.

**Figure 4.** The initial geometry used in the analysis and as starting point for the optimization. The surface mesh consists of 3 blades each with 36 square blocks. Each block has $32x32$ cells resulting in 110592 surface mesh cells.

## 3.2 Mesh convergence study

To quantify each solver's mesh dependency we list the integrated metrics, torque and thrust, for the three mesh levels L0, L1 and L2 in Table 4. As seen, a finer mesh level, L-1, was needed in order to determine if ADflow indeed was about to converge. This agrees with an earlier mesh convergence study (Dhert et al., 2017, Tab. 1) where up to $22 \cdot 10^6$ cells were used (without reaching convergence).

Table 4 clearly shows that the change from L0 to L-1 for ADflow is much less ($4\%$ in thrust and $7\%$ in torque) than the improvement from L2 to L1 ($15\%$ and $21\%$) or from L1 to L0 ($22\%$ and $41\%$). The error-columns make use of the Richardson extrapolation values from Fig. 5 to estimate the error in percentage from the 'true' continuum value estimated like:

$$f_{h=0} \approx f_1 + \frac{f_1 - f_2}{r^2 - 1}, \tag{7}$$



**Table 4.** Mesh convergence study for the compressible solver ADflow and the incompressible solver EllipSys3D. The operational conditions for the convergence study is wsp_08_L0 from Tab. 3.

| Mesh level | ADflow | | | | EllipSys3D | | | |
|---|---|---|---|---|---|---|---|---|
| | Thrust [N] | error † | Torque [Nm] | error ‡ | Thrust [N] | error †† | Torque [Nm] | error ‡‡ |
| L-1: $47.776 \cdot 10^6$ cells | $603 \cdot 10^3$ | 2.4 % | $4.547 \cdot 10^6$ | 2.2 % | $577 \cdot 10^3$ | 0.9 % | $4.471 \cdot 10^6$ | 0.2 % |
| L 0: $14.155 \cdot 10^6$ cells | $625 \cdot 10^3$ | 6.1 % | $4.877 \cdot 10^6$ | 9.6 % | $573 \cdot 10^3$ | 0.2 % | $4.457 \cdot 10^6$ | 0.5 % |
| L 1: $1.769 \cdot 10^6$ cells | $733 \cdot 10^3$ | 24.4 % | $6.156 \cdot 10^6$ | 38.3 % | $578 \cdot 10^3$ | 1.0 % | $4.402 \cdot 10^6$ | 1.7 % |
| L 2: $0.221 \cdot 10^6$ cells | $934 \cdot 10^3$ | 58.6 % | $10.403 \cdot 10^6$ | 134.5 % | $584 \cdot 10^3$ | 2.1 % | $4.336 \cdot 10^6$ | 3.2 % |

Error percentages calculated using Richardson extrapolations (from Fig. 5): † $5.89 \cdot 10^5$ [$N$], ‡ $4.45 \cdot 10^6$ [$Nm$], †† $5.72 \cdot 10^5$ [$N$], ‡‡ $4.48 \cdot 10^6$ [$Nm$]

where $f_{h=0}$ is the continuum value, $f_1$ and $f_2$ the value obtained on L0 and L1 respectively and $r$ is the grid refinement ratio. We refer to NASA's tutorial on spatial convergence [5] as well as the seminal paper by Roache (1994) for further details.

Besides quantifying the mesh dependency we identify the reassuring trait that the solvers seem to converge towards the *same* values. This, at least is true for the torque (less than 0.7%), whereas the continuum values for thrust seem to differ slightly more (3%). The table also allows us to estimate what mesh level should be adequate if we hope to obtain physically realistic

results. Here, the L0 is a reasonable compromise between accuracy (error $< 10\%$) and speed.

Looking at the error percentages in Tab. 4 we note the unexpected, slight increase in error for EllipSys3D in the thrust value on the finest mesh level. It also surprising that the compressible solver seems to benefit so drastically from increase in cell count but recent studies (Sørensen et al., 2016) have suggested that it can be the case for some compressible solvers albeit perhaps not as much as shown here. Looking at the Prandtl–Glauert (Glauert, 1928) lift and drag correction:

$$C_l^{comp} = \frac{C_l^{inc}}{\sqrt{1 - M^2}} \quad \text{and} \quad C_d^{comp} = \frac{C_d^{inc}}{\sqrt{1 - M^2}}, \tag{8}$$

where $M^2$ is the Mach number it is however not surprising that the compressible values are the larger ones. Indeed, the discussion of compressible versus incompressible solvers for wind turbine applications is more relevant than ever as size increases, and findings in the AVATAR project[6] also show that "compressibility effects play a role on large wind turbines"

(Sørensen et al., 2017, p. 9). For a recent study into the aerodynamic effects of compressibility for wind turbines we refer to the work by Sørensen et al. (2018).

### 3.3 Integrated loads

Integrated loads in the form of thrust and torque have been computed for each simulation in Tab. 3 and are visualized in Fig. 6. As seen, the ADflow results are consistently higher than the EllipSys3D results. This trend does agree with the mentioned

Prandtl–Glauert correction as well as the tendency from Tab. 4 where all thrust and torque values from ADflow are above

---

[5]https://www.grc.nasa.gov/WWW/wind/valid/tutorial/spatconv.html

[6]http://www.eera-avatar.eu/publications-results-and-links/





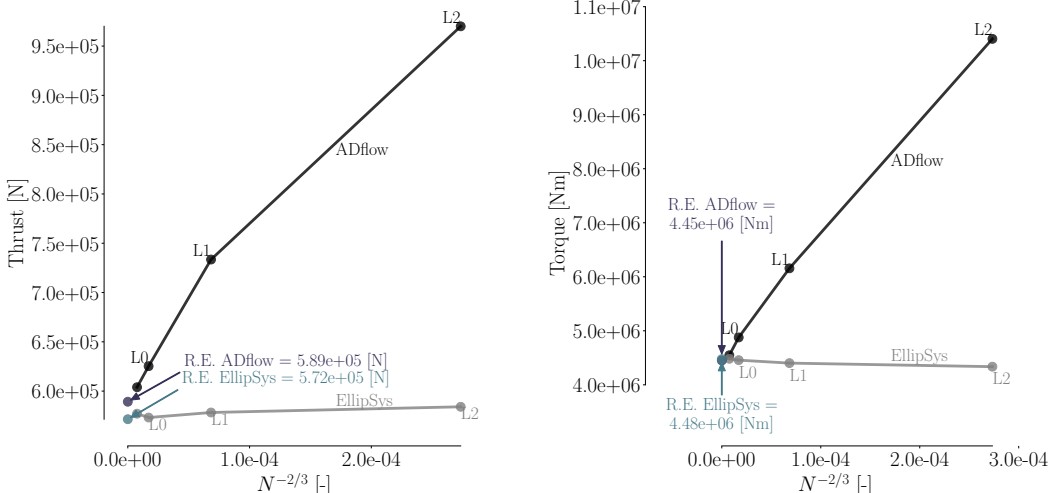

**Figure 5.** Richardson extrapolation found using Eq. (7) for the grid convergence study for thrust (LHS) and torque (RHS). Between the two solvers the extrapolated continuum values for thrust differ $3\%$ whereas the error for the torque values vary less than $0.7\%$.

their EllipSys3D counterparts. As a low-fidelity reference, we have added the integrated loads (in gray) from steady state BEM results using HAWCStab2. A general agreement between the CFD results and the HAWCStab2 results can be seen save perhaps for the torque value at $25\ m/s$, which could be corrected with a slight change in pitch setting given in Tab. 3. Agreement is expected between EllipSys3D and BEM since the airfoil data used in BEM is computed using EllipSys2D.

## 3.4 Spanwise forces, pressure distribution and flow visualization

Figure 7 shows the spanwise forces showing that the difference between solvers is more or less spread out over the entire span. Not surprisingly, the ADflow values are consistently higher. We will re-visit the distribution of spanwise forces after the optimization to inspect where performance increase occurs on the blade. Turning to the surface restricted streamlines in Fig. 8 we first note the rather large amount of separation. Even the pressure side shows a distinct area of separation from $19m$ to $41m$ span. Comparing said area with the pressure side separation for the unperturbed DTU 10 MW rotor in Fig. 9 where only a small separation area at the root is seen it is clear that the perturbed design we use as a starting point for the optimization seen in Fig. 8 suffers from thicker blades owed to the reduced chord distribution and increase in relative thickness. The suction side

in Fig. 8 looks more as one would expect save for the expanded separation area reaching just above $37m$ in spanwise direction. Here, the DTU 10 MW only has separation below $32m$ span as seen in Fig. 9.

In Fig. 10 we compare the obtained $C_p$ curves at three spanwise positions, namely; $35\ m$, $64\ m$ and $84\ m$ (positions marked in red in Fig. 8) where the $C_p$ distribution is found using the dynamic pressure and the farfield pressure, $p_\infty$:

$$C_p = \frac{p - p_\infty}{(1/2)\rho(V_\infty^2 + (r\omega)^2)}. \tag{9}$$

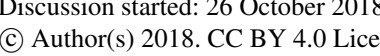



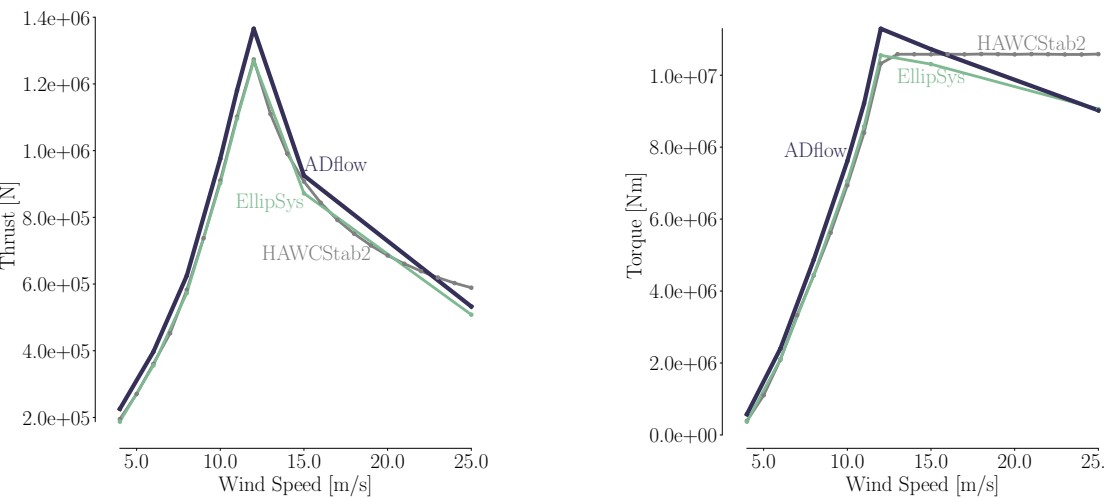

**Figure 6.** Total thrust (LHS) and torque (RHS) as a function of wind speed for the rotor geometry used as the starting point for the optimization. As expected, the torque increases rapidly from cut-in speed to the rated speed at 13 $m/s$, which is also where the thrust peak occurs. From rated to cut-out the torque curve flattens. Here, the pitch setting found with steady state BEM results using HAWCStab2 (seen in gray) clearly does not result in a completely flat line for the CFD solvers due to the model change. ADflow consistently overshoots the EllipSys results, which is consistent with the trend seen in Tab. 4. Operational conditions for the eight simulations are given in Tab. 3.

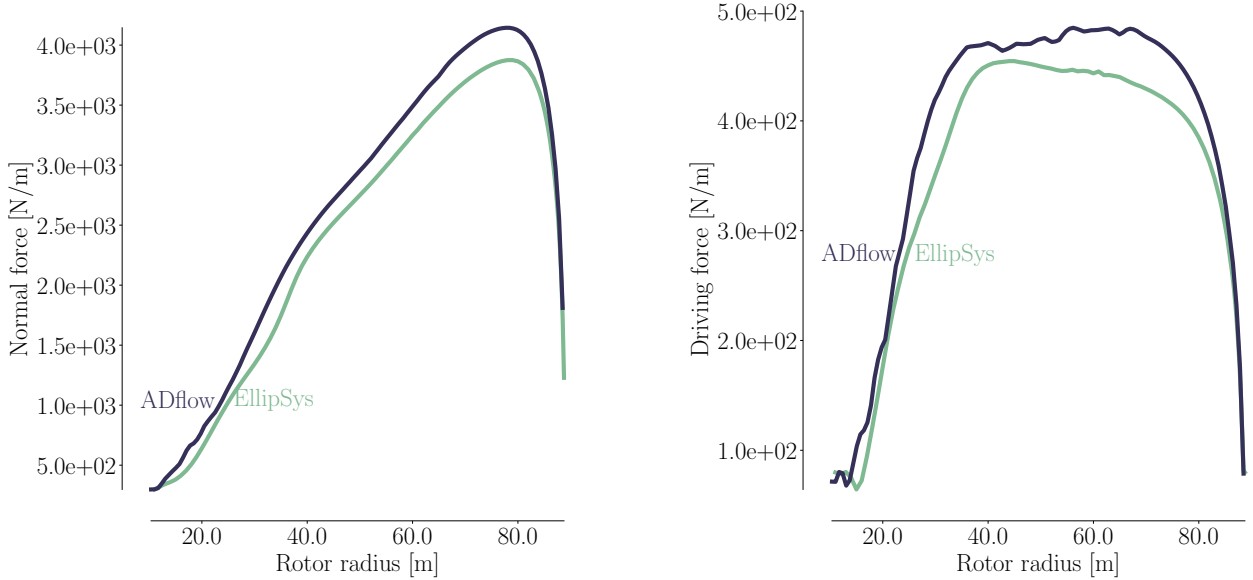

**Figure 7.** Spanwise distribution of the normal force (LHS) and driving force (RHS) for the operational conditions called wsp08_L0 in Tab. 3.

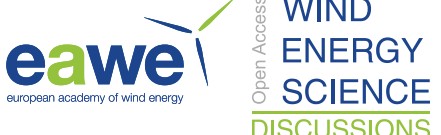

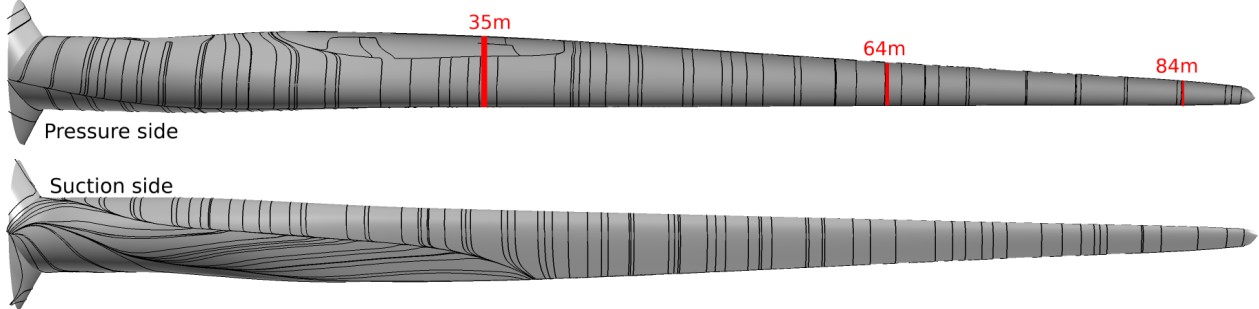

**Figure 8.** Surface restricted streamlines from the EllipSys solution both for the pressure side (top) and the suction side (bottom) for the perturbed design we use as a starting point for the optimization. The operational conditions are listed as the wsp08_L0 run in Tab. 3.

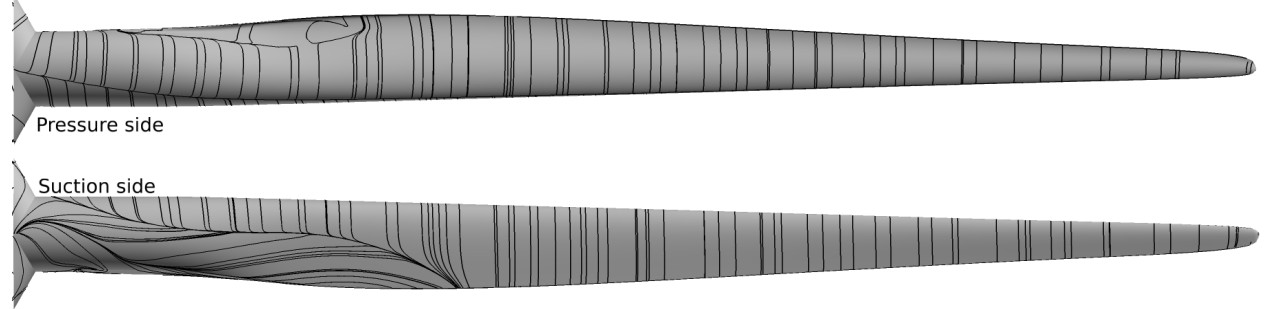

**Figure 9.** Surface restricted streamlines from the EllipSys solution obtained using the original DTU 10 MW wind turbine geometry for the wsp08_L0 run in Tab. 3 both for the pressure side (top) and the suction side (bottom).

The slice at $35\ m$ shows the least consistent comparison, which we suspect is due to the large amount of separation present both at suction and pressure side. Given that the solvers use different turbulence models it would be surprising to find a perfect match at this position. We also note that the pressure side separation results in a $C_p$ curve with a typical flat, squeezed shape in the $30\%$ closest to the trailing edge (TE). The $C_p$ curves for the sections at 64 m span and 84 m span show in general a better likeness of one another. Early investigations showed that the chordwise distribution of cells have a distinct impact on

the solvers' ability to capture the stagnation point and suction peak. Therefore, we chose a distribution that seemed to have enough cells close to the stagnation point while still having an adequate amount of cells to resolve the TE area. In general, the ADflow suction peaks seem to be more pronounced than those from EllipSys3D. The same can be said for the blunt TE, where the ADflow $C_p$ curve again has a more pronounced spike.

Having inspected various elements of the obtained solutions we conclude that while there certainly are discrepancies due

to difference in turbulence model, compressibility and stencil order to name but a few the overall picture rendered by the two solvers seem to resonate.




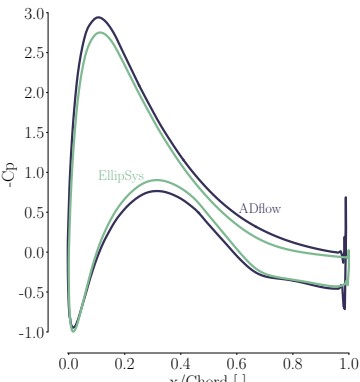
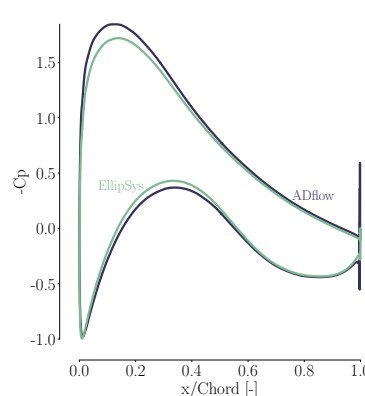
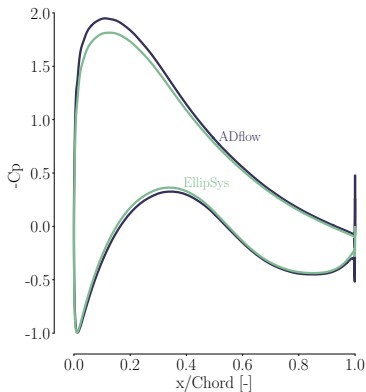

**Figure 10.** $C_p$-curves for 35 m (left), 64 $m$ (middle) and 84 $m$ (right).

## 4 Design Optimization Problem

The design optimization problem for the IEA Task 37 case study is to optimize AEP varying chord and twist while constraints on maximum increase in thrust of $14\%$ and maximum increase in bending moment, $M_b$, of $11\%$ must be satisfied. There is also a minimum absolute thickness on the inner $80\%$ span to ensure enough structural integrity. Mathematically, the design optimization problem takes the form:

$$\underset{x}{\text{maximize}} \quad AEP(x)$$

with respect to    $\text{Chord}_i, \ i = 1, \ldots, (\text{n} - 2)$

$$\text{Twist}_i, \ i = 1, \ldots, (\text{n} - 2)$$

$$\text{Shape}_i, \ i = 1, \ldots, \text{l} \cdot (\text{n} - 2)$$

subject to    $Thrust(x) \leq 1.14 \cdot Thrust(x_0)$

$$M_b(x) \leq 1.11 \cdot M_b(x_0)$$

$$Thickness(x) \geq Thickness_{IEA\ Task\ 37\ limit},$$

where $n = 9$ and $l = 10$ are resolution parameters for the FFD parameterization further detailed in the following section where

5   also the absolute minimum thickness constraint is visualized. Finally, we note that for all single point optimizations, i.e. when optimizing for a single wind speed, it suffices to optimize the total torque on the turbine. For multipoint optimizations the power is taken as the product of rotation rate and torque, $P = w \cdot Torque$. One can then use the Weibull distribution with scale and shape parameters $A = 8$ m/s and $k = 2$ to obtain a measure for the AEP. For multipoint optimization we use three wind speeds, namely; $5\ m/s$, $8\ m/s$ and $11\ m/s$, which approximately covers the range from cut-in to rated wind speed. The single

10   point optimizations are all carried out at a wind speed of $8\ m/s$ using the operational conditions listed in Tab. 3.





## 4.1 Parameterization

The initial WT design is seen in the LHS of Fig. 11 along with three FFD boxes with the resolution $(l, m, n) = (10, 2, 9)$ signifying 10 FFD points from leading edge (LE) to trailing edge (TE) and 9 sections in the spanwise direction. The FFD boxes each have two fixed, closely set sections at the root ensuring $C^1$ continuity and the seven outer remaining sections are all free to move and deform the blade. Several FFD box resolutions were tested and we found this box resolution to work best across all used mesh levels. Additional coloring on the blade pointing in the direction along the y-axis highlights the thickness constraints (blue) and the LE-TE-constraints (red). These constraints can be further inspected on the RHS of Fig. 11 where the curve plot shows the absolute thickness constraints that ensure a realistic amount of structural enforcement. These thickness constraints only cover the inner $80\%$ of the blade as detailed in the definition of the IEA case study. Below the curves a cross section of the FFD box is shown. The LE-TE constraint force the two, red LE FFD points to move exactly the same amount and in opposite direction. The same is true for the TE FFD points (also shown in red). LE-TE constraints ensure that the FFD box does not impose a skewing twist but only locally shape the profile. Two important aspects should be addressed with

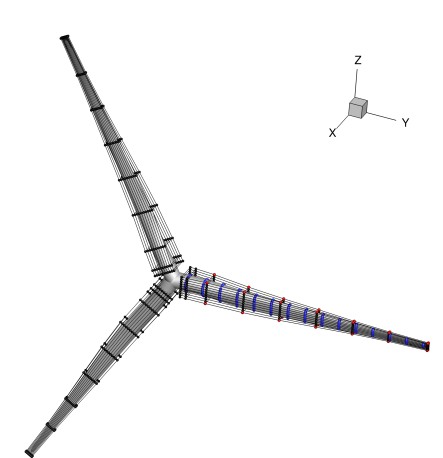

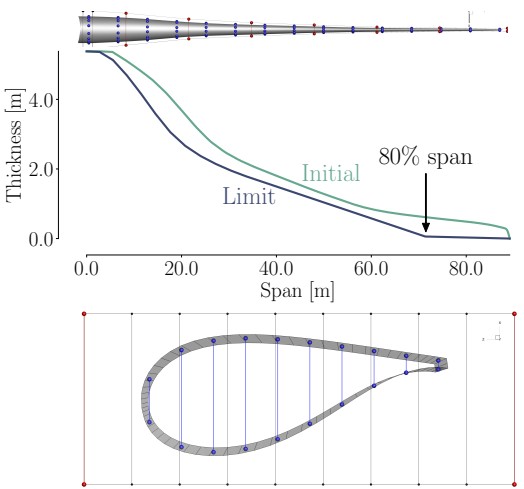

**Figure 11.** LHS: Overview of initial geometry and FFD boxes where the 9 spanwise sections are positioned at: $4.0\ m$, $6.0\ m$, $12.7\ m$, $25.5\ m$, $38.2\ m$, $51.0\ m$, $63.7\ m$, $76.4\ m$ and $90.2\ m$ span. RHS top: The blade pointing in the direction along the y-axis with 15 sections of thickness constraints (blue), 7 LE-constraints (red) and 7 TE-constraints (red). RHS middle: curve plot of initial (green) and minimum allowed absolute thickness (purple) covering $80\%$ of the span. RHS bottom: profile section at span $= 36$ m showing how well the 10 chordwise thickness constraints (blue) cover an airfoil.

respect to the thickness constraints. Both aspects are deviations from the described IEA case study and will therefore be clearly stated below: First of all, the thickness limit curve seen in the graph on the RHS of Fig. 11 was imposed exactly as seen for the planform optimization but for the single and multipoint optimizations the limit was imposed only for the fourth thickness constraint counting from the LE. Thickness constraint 3 and 5 counting from the leading edge were relaxed to 85 % of the





full constraint at the given spanwise position. This was done in order to ensure space for structural enforcement on one hand while on the other hand to allow the optimizer as much freedom as possible. Finally, we set the eighth and the tenth thickness constraint counting from the LE to 15% and 90% of the full thickness constraint respectively. All other thickness constraints per section were removed so that the FFD boxes could deform the airfoils into novel shapes as freely as possible.

The second item to address is a change in thickness constraints for the outer 20 % of the span. Where the previous change

was a relaxation of the IEA constraint, this is not the case for the second change. Since the case study from IEA task 37 simply does not give any minimum thickness constraints on the outer most 20 % of the span we found that the optimizer shaped the blade into a sword-like object where the thickness collapsed to a paper thin object. This is obviously not an industrially relevant shape and thus we imposed the additional (minimum) thickness constraint that the outer most 20 % of the blade should have a relative thickness equal to that at 60 m span for the initial shape. This ensured that the optimizer could still reduce the thickness

at the tip without collapsing the thickness altogether.

## 5   Results

The results are split into four parts, which we briefly describe and subsequently summarize in Tab. 5: First we show a single design variable optimization where pitch is varied to optimize torque. It is included to validate the basic adjoint formulation for rotating frame of reference flows. Secondly, we show a planform optimization where chord and twist are varied. This

optimization is well suited to compare with BEM results since the FFD boxes here have not been allowed to locally deform the blade. The two final optimizations are the single and multipoint optimization where the free form deformation is allowed to locally re-shape the airfoils and change twist and chord distributions.

**Table 5.** Overview of all optimizations.

| Optimization | Objective | Design variables | |
| --- | --- | --- | --- |
| Pitch | Torque | 1 | |
| Planform | Torque | 14 | (BEM comparison) |
| Single point | Torque | 154 | |
| Multipoint | AEP | 154 | |

### 5.1   Pitch optimization

In the pitch optimization the pitch angle for the seven outer FFD sections on each blade is controlled by a single design

5   variable. The result is an increase in torque of 20.2%, 18.7% and 16.3% for mesh level L2, L1 and L0 respectively. Figures 12-14 summarize the optimization history for the three mesh levels. Fig. 12 shows the steps taken by the optimizer for each mesh level. Up to 20 iterations were needed to converge. Black lines symbolize parameter sweeps and colored lines show the actual steps taken by the optimizer. Parameter sweeps are of course a luxury one can only afford for (very) few design variables given that the flow field must be converged for each evaluated point. However, the sweeps do allow for an intuitive





10 'sanity' check and indeed it seems all $x^*$ are true optima. Although the torque value varies greatly between mesh levels we see consistency in the results with the consensus being around 7 degrees and only a few degrees spread. Also a fourth color (light purple) can be seen. This optimization uses a hotstart meaning that the result from the coarse mesh level (L2) is used on the finest mesh level (L0) and consequently only a few steps have to be taken on the finest mesh before convergence is attained. The result obtained on the finest mesh level from scratch is reassuringly identical to the result from the hotstart simulation.

Turning to Fig. 13 we find that all optimizations have converged to minimum $10 \cdot 10^{-5}$ (black, dashed line). Indeed, we have observed convergence down to $10 \cdot 10^{-9}$ for several cases but for all purposes a threshold of $10 \cdot 10^{-5}$ seems to be adequate and marks where the majority of change already has occurred. This can be observed in Fig. 14 where all curves (save for the hotstart) raises steeply at the beginning but flattens out towards the end. Clearly, one does not gain much by converging

5 the problems further. Finally, we note that the `Merit Function` in SNOPT is an augmented Lagrangian merit function as detailed in the SNOPT manual. This means that for unconstrained problems as in the present pitch optimization the merit function is equal to the objective function, which is torque in this optimization. When constraints are present the merit function does not necessarily equal the objective function. However, as we approach the solution it will converge to the objective function. As seen in Fig. 14 the values are furthermore scaled to be close to unity to get a better performance from SNOPT. Table **??** summarizes the pitch optimizations.

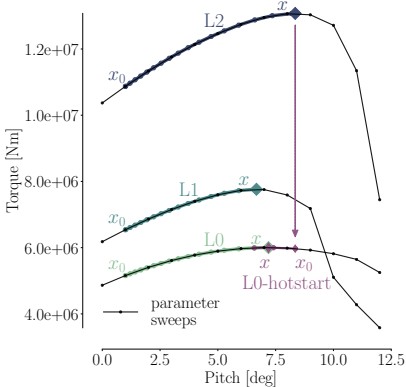

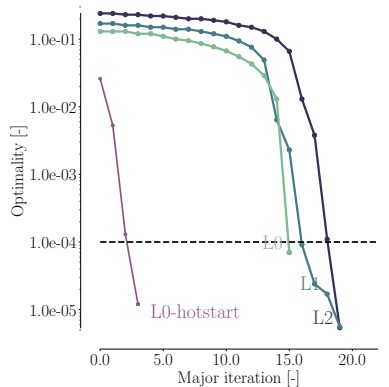

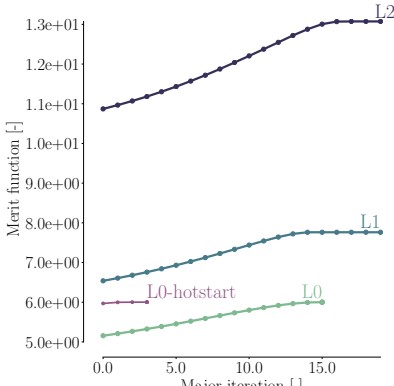

**Figure 12.** A visualization of the steps taken by the optimizer to converge the single design variable pitch optimization for mesh level L2, L1 and L0. Also a hotstart optimization can be seen, which uses the result from a coarse mesh (L2) as the starting point on the finest mesh (L0) resulting in much fewer steps but an identical result as can be verified in Tab. **??**.

**Figure 13.** Convergence history for the pitch optimizations. Where the three normal optimizations (L2, L1 and L0) starting from scratch clearly need close to 20 steps before convergence occurs we note that the hotstart seems closer to the basin-of-attraction and converges in 4 steps.

**Figure 14.** Merit function history as a function of steps taken by the optimizer. From Fig. 13 one can deduct that once converged below $10 \cdot 10^{-5}$ curves flatten and further convergence seems futile.



**Table 6.** Optimization with one pitch design variable. The increase in objective function, torque, is 20.2%, 18.7% and 16.3% for mesh level L2, L1 and L0 respectively.

| Mesh | Solution $x^*$ | Torque( $x^0$) | Torque( $x^*$) | CPU time | Iterations | Optimality |
|---|---|---|---|---|---|---|
| | [deg] | [Nm] | [Nm] | [h] | [−] | [−] |
| L 0: $14.155 \cdot 10^6$ cells | 7.19 | $4.88 \cdot 10^6$ | $6.00 \cdot 10^6$ | 12734.7 | 16 | $7.0 \cdot 10^{-5}$ |
| L 0:(using $x^*_{L2}$ hotstart) | 7.19 | $4.88 \cdot 10^6$ | $6.00 \cdot 10^6$ | 6436.3 | 4 | $1.2 \cdot 10^{-5}$ |
| L 1: $1.769 \cdot 10^6$ cells | 6.67 | $6.12 \cdot 10^6$ | $7.76 \cdot 10^6$ | 1004.1 | 19 | $1.7 \cdot 10^{-5}$ |
| L 2: $0.221 \cdot 10^6$ cells | 8.35 | $10.40 \cdot 10^6$ | $13.07 \cdot 10^6$ | 106.9 | 20 | $5.4 \cdot 10^{-6}$ |

### 5.2 Planform optimization

For the planform optimization both twist and chord can be adjusted at the seven outer FFD sections along the blade, which results in 14 design variables. As mentioned, the optimizer cannot impose a skewing twist through the local shape variables and thus, the twist variable should be comparable to the twist variable utilized in BEM codes. For the chord variable there is a slight discrepancy in problem definition compared to the BEM codes since they at each section are allowed to blend between a select few number of airfoils ranging from 72% to 24% in relative thickness. For our setup we maintain the relative thickness at each section by scaling both chord and thickness with the same variable. Thus we avoid a degeneracy once the local shape variables are activated. However, the chord length is free to be scaled just as for the BEM codes and and the exact same constraint in absolute (minimum) thickness is satisfied. Due to the above design variable discrepancy we will show two BEM results: One optimization from a single wind speed of 8 m/s without the relative thickness design variable named BEM1. This optimization should be as close to the presented CFD-based planform optimization as possible. The other BEM optimization result we show is for wind speeds $5\ m/s, 8\ m/s$ and $11\ m/s$ where the relative thickness is allowed to vary as explained above. We use the SNOPT optimizer for all BEM optimizations just as we do for the high-fidelity results.

The high-fidelity planform optimization results are visualized in Fig. 15-17 showing final chord and twist distributions as well as history of convergence and merit functions. Inspecting the upper plot in Fig. 15 we find that the optimized shape for the finest mesh level has a large increase in chord towards the root and a slimmer blade towards the tip just as we would expect for an aerodynamically optimized blade. The optimized chord distribution is reminiscent of the DTU 10 MW turbine's chord distribution from the LHS of Fig. 2, which was also designed for maximum $C_p$. However, a difference towards the root is clearly discernible where the DTU 10 MW chord distribution hardly reaches more than $6\ m$, which is due to a geometric constraint on maximum chord of $6.2\ m$. The optimized twist seen as the green curve in the lower plot in Fig. 15 exhibits a more aggressive distribution towards the tip compared to its baseline. Interestingly, there seems to be a small, unexpected increase in twist between 20-30 m span for the L1 mesh. A trend that disappears again for the L0 mesh, which better resembles the DTU 10 MW twist distribution seen in the RHS of Fig. 2.



Looking at the result across mesh levels one finds a much larger spread than for the pitch optimization. Clearly, the result from L2 bears little resemblance to that from L0. The results from L2 are particularly disturbing since it actually seems to highlight the exact opposite trends with a slightly slimmer blade towards the root and a minor increase in chord between 40-70 m span. The counterintuitive performance of the L2 mesh is not a surprise considering the large error in the solution at this grid level seen in the grid dependence study. Interestingly, the amount of difference in chord distribution from L1 to L0 also seem more noticeable than warranted by a previous mesh convergence study (Lyu et al., 2013, Fig. 8) albeit this study was carried
5   out on a previous version of the CFD solver. However, save for the chord increase from L1 to L0 towards the root both twist and chord distribution seem fairly consistent and one could still obtain valuable design insight while saving time by using the L1 mesh.

Fig. 16 shows convergence history for the three mesh levels. Again, all optimizations were converged down to at least $10 \cdot 10^{-5}$ marked as a black, dashed line. In Fig. 17 we find a similar trend as seen for the pitch optimization in Fig. 14 namely,
10   that much of the improvement is gained in the first half of the optimization. Thus, an easy way to speed up the design process could simply be to take an intermediate design. However, one should make sure to check the feasibility since SQP methods easily can explore infeasible regions temporarily.

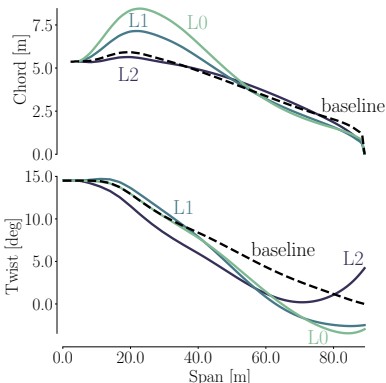

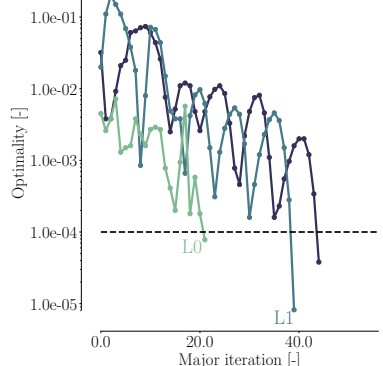

**Figure 15.** Final chord and twist distributions for the planform optimization using the operational conditions called 'wsp08_L0' in Tab. 3. Initial design is seen as a black, dashed curve. Note that the result on the coarse mesh (L2) actually suggests trends, which are in the *wrong* direction with a slightly reduced chord at the root and an increase in chord further out along the span.

**Figure 16.** Convergence history for all three mesh levels. Similar to the pitch optimization (Fig. 13) we set the threshold to $10 \cdot 10^{-5}$ and all simulations have been converged below this level.

**Figure 17.** Scaled merit function history for the planform optimization showing the majority of increase comes during the first half of the optimization. The sharp initial decrease for L1 is due to the (infeasible) hot-start from L2. Where the merit function was exactly as printed by SNOPT in Fig. 14 we have for the planform optimization normalized all three merit functions to the range [0, 1] since the scalings were too misaligned for visual comparison.




### 5.2.1 IEA case study comparison

We now compare the L0 result from the planform optimization to the BEM results obtained with HAWTOpt2 Zahle et al. (2016) that interfaces to HAWCStab2 Hansen (2004). Since it is a comparison across fidelities with completely different models we do not expect a one-to-one match but hope to find similar trends in the optimized designs. Visualization of chord (LHS) and twist (RHS) distributions can be seen in Fig. 18.

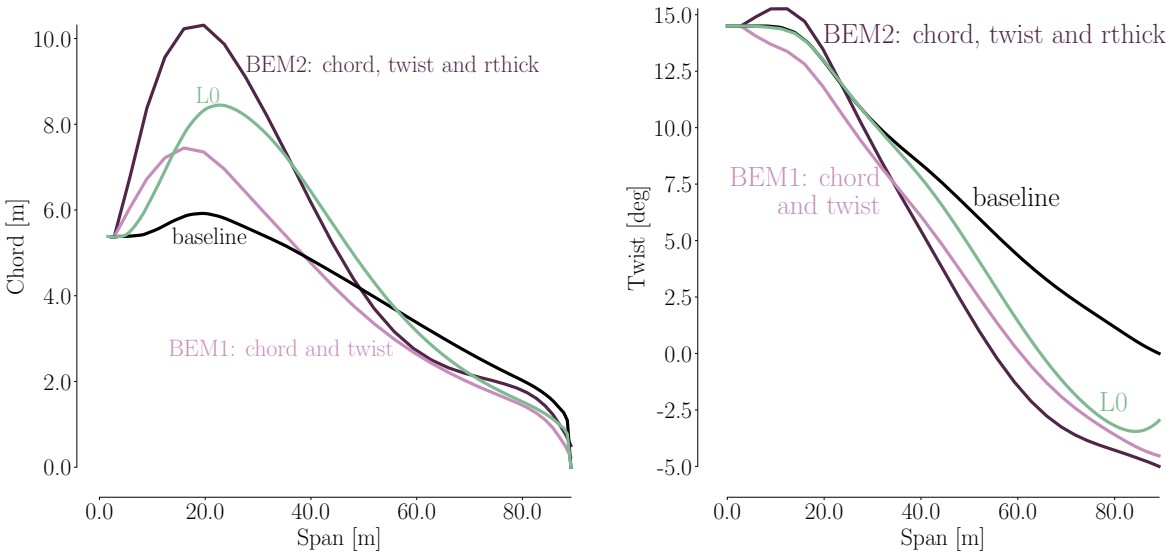

**Figure 18.** Comparison across fidelities for both chord (LHS) and twist (RHS) where the L0 result (green) from the high-fidelity planform optimization is seen along side 2 BEM results from the same IEA task 37 case study. The light purple BEM1 result is for a wind speed of 8 m/s where only chord and twist design variables are enabled. The darker purple BEM2 result also has the freedom to vary relative thickness and is over multiple wind speeds, namely: $5\ m/s, 8\ m/s$ and $11\ m/s$. The former BEM result should be very similar to the CFD-based result in problem definition.

Although both chord and twist distributions show clear discrepancies for the final designs, there does seem to be several traits in common. Inspecting the chord distributions we find a rather large difference in suggested increase in max chord. Here, BEM1 suggests 26 % increase whereas BEM2 suggests 74 % increase. The CFD-based result lies in the middle with 43 %. BEM1 is the surprising result of the three where it seems the relation between power and thrust is so poor that it makes little sense to increase the chord at the root. By adding the relative thickness as a variable it makes sense that BEM2 suddenly can increase the chord further. Given that our CFD-based optimization has fixed relative thicknesses it also makes sense that BEM2 has larger chord values. The BEM optimizations use Bezier splines where the design variables are positioned with the same spanwise distribution as the FFD sections. However, the results have a steeper, more pronounced increase, which we



suspect our CFD setup could not imitate even if it wanted to. The two innermost locked FFD sections (positioned at 4m and 6m span) ensuring the $C^1$ mesh continuity make such a steep increase in chord impossible so close to the root. It is actually quite impressive that the deformation routines can even stretch the maximum chord 43 %. In fact, an earlier study (Dhert, 2015, p. 58) completely ruled out design variables such as chord in fear of mesh failure. However, it would be very hard to maintain a high grid quality with a deformation similar to the BEM2 solution and avoid negative cell volumes due to a similar expansive

deformation from neighboring blades. As a final comment on the discrepancies at the root we note that BEM profile data for such thick airfoils most likely are far from precise. Besides, the empirical 3D correction used on said 2D profile data is also likely to be imprecise. Needles to say, the combination of the two could yield shaky results. To make matters worse, we know from the comparative analysis (Fig. 8) that separation reaches up to about 37 m span, which further muddies the picture. A more unison picture is seen for the tip region where the chord distributions have converged to a reduced chord consensus where

only minor differences can be seen.

As for the twist variable in the RHS of Fig. 18 the overall trend of a more aggressive distribution seem evident between the BEM1 result and the CFD result albeit the former consistently lies 1-2 degrees below the latter. We have no immediate explanation for this misalignment although we note that the twist distribution for L0 perhaps cannot descend as close to the root as the BEM results because of the aforementioned 2 locked FFD sections although further investigations are needed to fully clarify the matter. The result from BEM2 is the lone distribution suggesting an initial increase in twist before the familiar

trend of a more aggressive twist is seen.

Looking at the bigger picture it is noteworthy that albeit differences certainly are present we find the same major trends in final design obtained with vastly different models, namely BEM on one hand and RANS CFD on the other. Reassuringly, the overall trends mimic those from the unperturbed 10 MW RWT design in Fig. 2. For completeness we have summarized the outcome of the planform optimization as well as the BEM results in Tab. 7, where we note that although the different

mesh levels do result in somewhat similar amounts of improvement it is clear from Fig. 15 that different mesh levels result in (very) different designs. One should remember when comparing the $11.07\%$ improvement for L0 with the $8.06\%$ and $22.46\%$ improvement from BEM1 and BEM2 that BEM2 is improvement in AEP using the Weibull distribution whereas the other results simply are torque improvements.

Using values for torque from Tab. 7 we can obtain the power coefficient, $C_P$, defined as:

$$C_P = \frac{P}{(1/2)\rho V^3 A} \tag{10}$$

where P is power, $\rho$ is density, V is wind speed and A is swept area. The resulting coefficients are $C_P$: 1.04, 0.62 and 0.48 for mesh levels L2, L1 and L0 respectively. Clearly the coarser the mesh the more unphysical the coefficient. The Betz limit

for power coefficients, $C_{P\ Betz} = 0.59$, is evidently violated for L2 and L1, which draws the results from coarse mesh levels into doubt. Judging from the huge spread in these coefficients it is not surprising that the optimized design differ greatly across mesh levels.





**Table 7.** A comparison to BEM results from IEA Task 37.

| Mesh level | ——————- present work † ——————- | | | ————— IEA Task 37 ————— | |
| | Torque( $x^0$ ) [$Nm$] | Torque( $x^*$ ) [$Nm$] | Improvement | BEM1 †† Improvement | BEM2 ††† Improvement |
|---|---|---|---|---|---|
| L 0: $14.155 \cdot 10^6$ $cells$ | $4.88 \cdot 10^6$ | $5.42 \cdot 10^6$ | 11.07% | | |
| L 1: $1.769 \cdot 10^6$ $cells$ | $6.12 \cdot 10^6$ | $6.88 \cdot 10^6$ | 12.42% | 8.06% | 22.46% |
| L 2: $0.221 \cdot 10^6$ $cells$ | $10.40 \cdot 10^6$ | $11.57 \cdot 10^6$ | 11.25% | | |

† The relative improvement percentage is here the improvement in torque since the high-fidelity results stem from a single point optimization †† The relative improvement percentages from BEM1 is the improvement in torque using wind speed: 8 [m/s] † † † The relative improvement percentages from BEM2 is the improvement in AEP where wind speeds 5 [m/s], 8 [m/s] and 11 [m/s] have been considered using a Weibull distribution.

### 5.3 Single point shape optimization

We now allow all FFD points (save for the two locked root sections) to locally deform the blade shape. Fig. 19 shows conver-
gence (LHS) and scaled merit function (RHS) history for the free form shape optimizations on all three grid levels. Comparing
the convergence history to similar plots for pitch and planform optimizations (Figs. 13 and 16) it is noticeable that only on the
finest mesh level can the threshold (black, dashed line) approximately be met. However, the scaled merit function plots on the
RHS do seem flat for L2 and L1 (albeit the latter curve is less smooth) hinting that the merit function could have plateaued.

Table 8 shows the gained improvement for each mesh level. As a first sanity check one can verify that no results are below
the improvements obtained in the planform optimization. This makes complete sense, given that the optimizer now has more
freedom to deform while the same constraints are applied except for the slight change in thickness constraint for the outer
% of the blade as described in Sec. 4.1. Inspecting the results in Tab. 8 it seems that the improvement from the planform
optimization (Tab. 7) can be more or less doubled depending on the mesh level by allowing the optimizer to locally shape the
profiles. We ascribe this improved gain to lower relative thickness when we allow the optimizer to freely shape the profiles
instead of fixing the relative thickness. Finally, we note that the BEM2 improvement in Tab. 7 is comparable to the single point
results for two mesh levels. Where BEM1 is as close as possible to the planform optimization BEM2 is comparable to the
single point optimizations since relative thicknesses can change in both cases. It is therefore not unreasonable that the BEM1
result is closest to the planform results in Tab. 7 whereas the BEM2 results are closer to the single point results in Tab. 8.

It should be clearly stated that these results do *not* show that the industry necessarily can gain 20 % increase simply by using
high-fidelity optimization. Indeed, the improvement is somewhat arbitrary since we study an intentionally poor design subject
to equally arbitrary loads constraints.




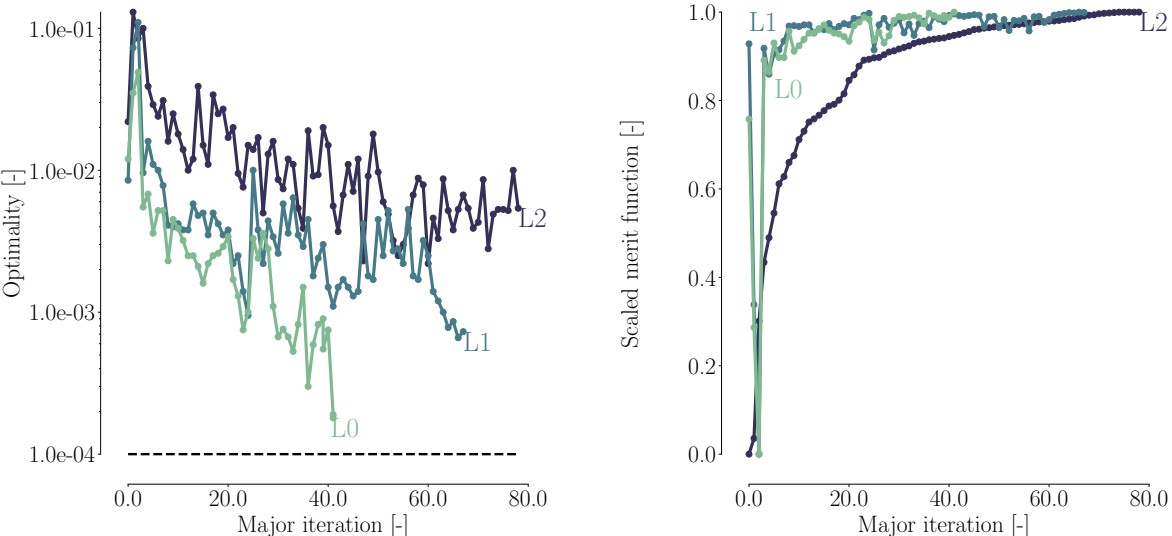

**Figure 19.** Convergence history (LHS) and (scaled) merit functions history (RHS) for the single point optimizations. The threshold is closest to being met on the finest mesh level. However, looking at the merit function history, for L2, for example, which has most difficulty gaining deep convergence one can easily verify that there is little gain in continued iterations as the scaled (purple) merit function is all but flat for the final iterations. The first few steps for L1 and L0 merit functions are usually quite fluctuating due to infeasible hotstarts from coarser grid levels.

**Table 8.** Overview of single point optimization results.

| Mesh level | Single point optimization † | | |
| --- | --- | --- | --- |
| | Torque($x^0$) | Torque($x^*$) | Improvement |
| | $[Nm]$ | $[Nm]$ | |
| L 0: $14.155 \cdot 10^6$ cells | $4.88 \cdot 10^6$ | $5.65 \cdot 10^6$ | 15.78% |
| L 1: $1.769 \cdot 10^6$ cells | $6.12 \cdot 10^6$ | $7.37 \cdot 10^6$ | 20.10% |
| L 2: $0.221 \cdot 10^6$ cells | $10.40 \cdot 10^6$ | $12.70 \cdot 10^6$ | 22.11% |

† For the wsp08_L0 operational conditions seen in Tab. 3





5   Turning to the actual optimized design we show a comparison to the baseline geometry in Fig. 20 where we have focused on the $C_p$ distribution. By simply comparing the baseline (top) and optimized design (just below) it is evident, that the optimized blade has an increase in chord close to the root as was hinted at already in the planform optimization.

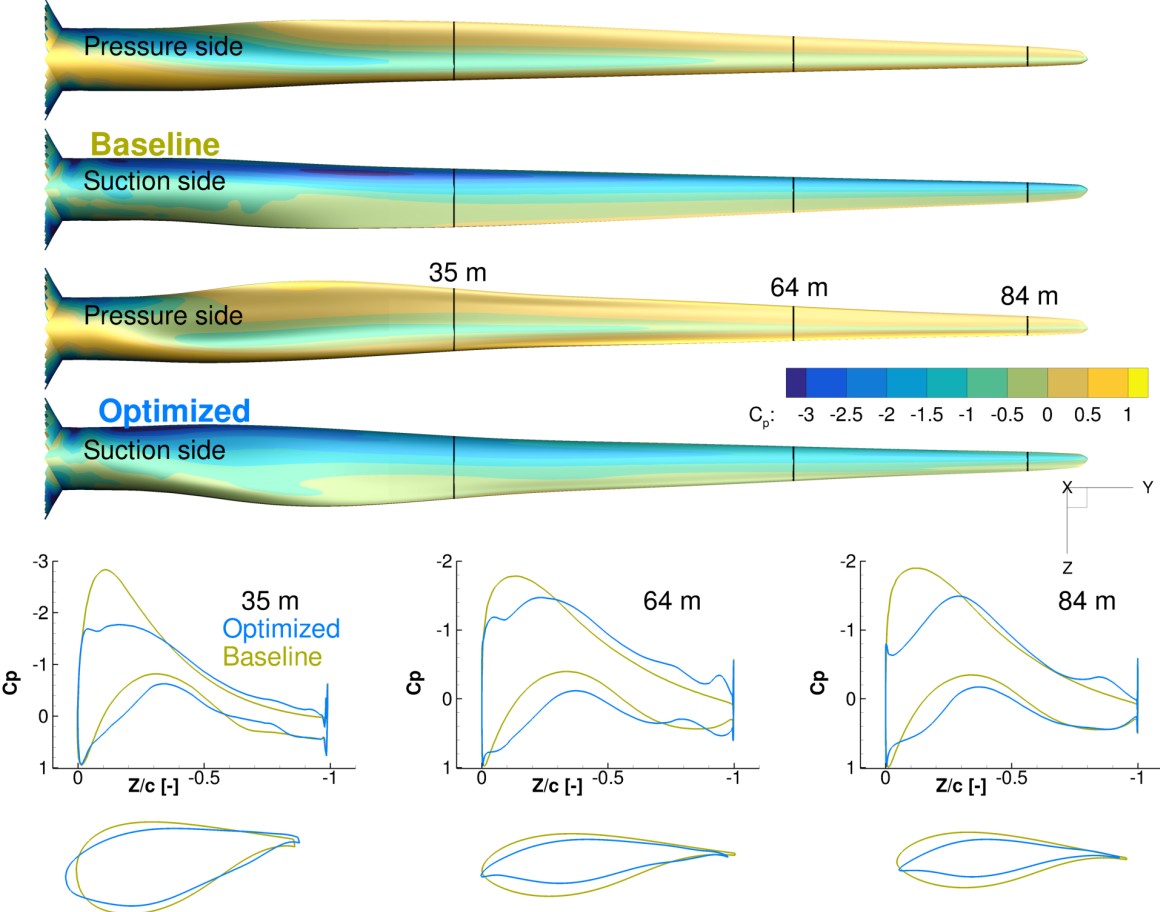

**Figure 20.** Comparison of $C_p$ curves for baseline and the single point optimized result. TEs, especially at the root, have an increase in camber and also LEs have changed resulting in less pronounced suction peaks.

Inspecting the slices at $35\ m$, $64\ m$, and $84\ m$ span we show their (scaled) profiles at the bottom of Fig. 20 and $C_p$ curves just above each profile. The new airfoils have a slight increase in camber and a clearly reduced relative thickness, which we expected since the perturbed initial design had rather large relative thicknesses. The increase in camber to generate more lift resonates with earlier findings (Dhert et al., 2017) where "extreme" cambering at TE was observed for airfoils near the root. However, utilized mesh levels are certainly not alike ($2.6 \cdot 10^6$ cells vs. $14.16 \cdot 10^6$ cells), which can lead to very different results as we have seen. We also noted already in the introduction, that the extreme cambering could be owed to not having





chord as a design variable. Since we do have chord as a variable we see a much more modest increase in camber. All leading edges also seem to have changed. Particularly at $64$ m and $84$ m span the LE has become more pointed. In relation to this change in LE one can notice that suction peaks have become less pronounced. The LE shape is clearly not desirable. In single point optimization we only take one angle of attack (AOA) into account and as a result, we should implement a curvature constraint to get better final designs. Another option is of course to do multipoint optimization. However, one must make sure that some of the multipoint wind speeds lie outside the constant tip speed ratio (TSR) range. For this reason we chose the wind speed, $5\ m/s$, which will force the optimizer to find an improved design while taking more than one AOA into account. In the multipoint section below, we demonstrate both approaches. First, we use three wind speeds in a multipoint optimization to investigate the effect of taking multiple AOAs into account. Then, we add a geometrical constraint on the LE and compare the profiles.

### 5.3.1 Performance of optimized design

To inspect the performance improvement we show spanwise forces for both optimized and baseline design in Fig. 21. The optimized rotor seems evenly loaded, which is typical for $C_p$-optimized rotors. To get a different distribution one could add span as design variable, which presumably would result in longer blades with a more triangular load distribution. This could certainly be an interesting avenue to explore in future works. The peak of the normal force for the optimized design in the LHS of Fig. 21 is further inboard suggesting that the flapwise moment constraint is driving this characteristic. We find that the constraint on flapwise bending moment results in a more moderate peak increase. It is our experience that this constraint is crucial to get realistic designs.

### 5.4 Multipoint shape optimization

Finally, we present results from multipoint optimizations for all mesh levels using the wind speeds $5\ m/s$, $8\ m/s$, and $11\ m/s$. The motivation for multipoint is to take a whole range of wind speeds into consideration thereby gaining a more robust design. As seen in the LHS of Fig. 6 it is at $12\ m/s$ where maximum thrust occurs and we will therefore use this value in the flapwise bending moment constraint. Importantly, this was also done for the IEA BEM2 results to better compare with the multipoint results. The design optimization problem and overall setup was detailed in Sec. 4 and the main difference to the single point optimizations is how the merit function is computed, namely by taking the Weibull distribution into account.

The history of convergence and merit function are shown in Fig. 22. Just as for the single point optimization, the selected threshold is not quite met. However, the scaled merit function seems to have flattened at least for L1 and L2.

As we pointed out above, we will force the optimizer to take a second AOA into account by including the $5\ m/s$ wind speed since it lies outside the constant TSR range. We therefore expect the problematic LE shape to improve and looking at Fig. 23 we do see the desired effect. The final design can, however, be further improved either through curvature constraints or by taking more than two AOAs into account. Finally, we point out that the suggested camber at the TE for the middle profile in Fig. 23 is not desirable since the flow would likely separate at the trailing edge for slightly higher AOAs than those taken into account.

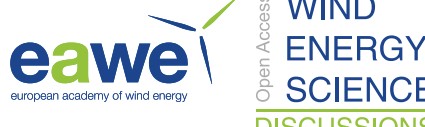



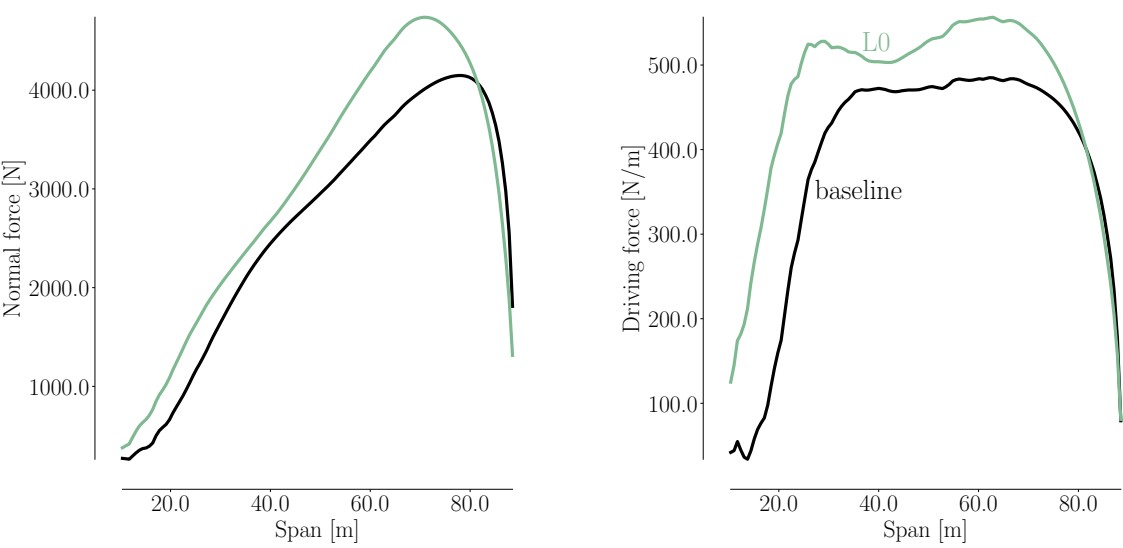

**Figure 21.** Comparison of normal (LHS) and driving (RHS) forces for baseline and optimized design. There is a moderate increase in normal thrust and the peak has also moved further inboard.

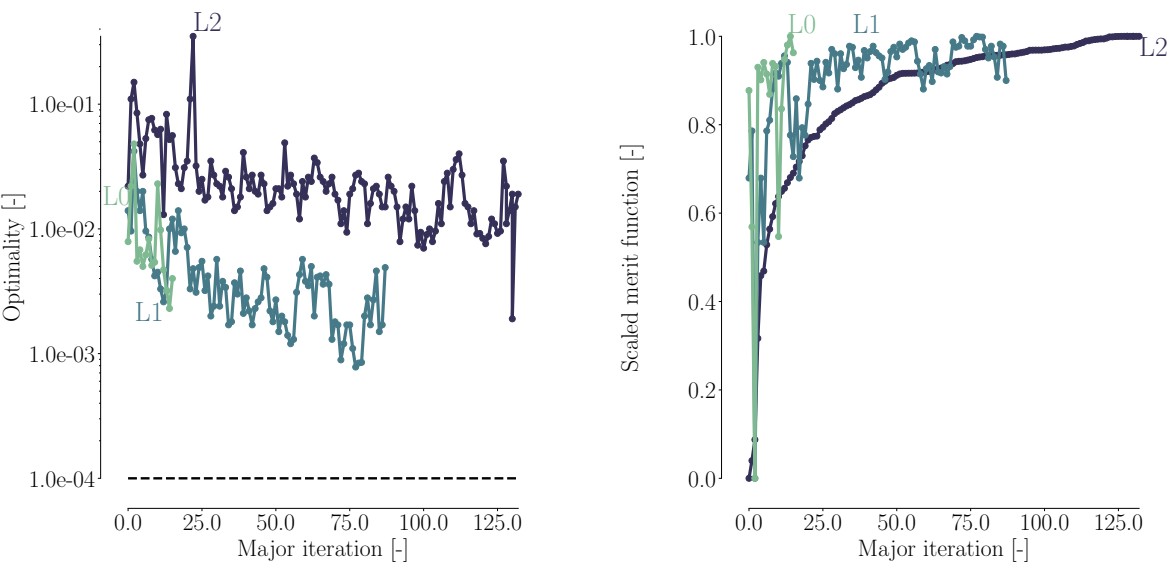

**Figure 22.** History of convergence (LHS) and scaled merit function (RHS) for the mulitpoint optimization.





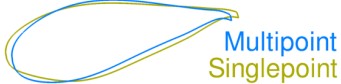
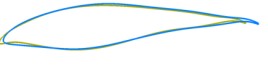
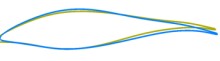

**Figure 23.** Comparison of profiles obtained from singlepoint and multipoint optimizations. The single point slices are from 35 $m$, 64 $m$, and 84 $m$ span and are also seen in Fig. 20.

To support the claim that a geometrical constraint improves the LE shape we add a LE constraint and re-run the multipoint optimization. The effect of adding the LE constraint can be seen in Fig. 24 where the previous multipoint results are compared to profile shapes obtained using the LE constraint. From the figure it is clear that the LE is much rounder. Ideally, one could render the geometrical LE constraint superfluos by including more AOAs but in reality, it is our experience that said constraint is a must. Inspecting the final design (blue) in Fig. 24, there are evidently other deficits to address. We limit this discussion to two aspects. The first improvement is to impose a geometrical constraint to counter the concave profile shape seen towards the TE on the suction side. The second notion is more grave and relates to the fluid modeling. Usually, the maximum thickness occurs closer to the 30 % chord region. However, the suggested shapes in Fig. 24 are rather nose-heavy due to the fully turbulent fluid modeling. A transition modeling should therefore be added which would result in a more smooth change in curvature from LE to TE and thereby a less nose-heavy profile shape. It is, in other words, evident, that although the single point design was improved by using multipoint optimization, some work still remains before industrially relevant designs can be achieved.

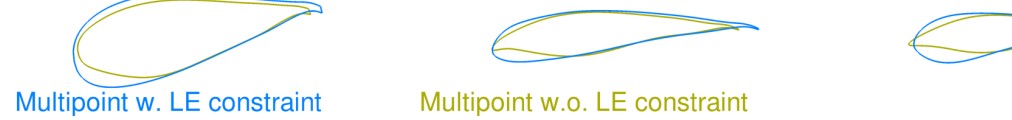

**Figure 24.** Comparison of profiles obtained from multipoint optimizations with (blue) and without (yellow) a geometrical constraint that prevents a colapse of the LE. Comparing the present figure with Fig. 23 it is clear that the geometrical constraint inclusion results in a much rounder LE.

As a concluding remark for the multipoint section we conclude that the multipoint setup is functional but should be further improved to obtain truly robust aerodynamically designed profiles. The improvement (except for the above mentioned notions) should result in a better control of minimum and maximum lift coefficient, $C_l$, which is known to be crucial for the resulting loads. Indeed, Madsen et al. (2014) have shown that the AOA varies quite a lot even for a wind speed within the constant TSR range. Ideally, one should therefore optimize performance within a range of AOAs by e.g. varying the rotation rate or pitch setting for a given wind speed.





## 6 Conclusions

In this work, we presented results from high-fidelity RANS-based shape optimization of a 10 MW RWT. Based on our literature review of the high-fidelity shape optimization efforts in wind turbine design, we determined that this was a promising area of research that has just begun. We made a comparative analysis between state-of-the-art compressible and incompressible CFD

solvers to quantify the mesh dependence and discrepancies across different RANS models applied on the same rotor. We then studied the advantage of using higher fidelity models by comparing our optimization results to low-fidelity BEM results from the same case study. We did this through a planform optimization with chord and twist variables where shape changes were restricted to keep the design case comparable with the BEM-based optimization. Overall design trends were recognized across fidelities, while certain differences partly due to choice in parameterization were observed. The same overall amount

of improvement was observed, which was approximately doubled for full shape optimization entailing twist, chord, and shape design variables which raised the number of design variables from 14 to 154. The increase in improvement was brought about by a decrease in relative thickness as well as the novel airfoil shapes. The present demonstration shows that with the right tools, we can model the entire geometry, including the root, and optimize modern wind turbine rotors at the cost of $\mathcal{O}(10^2)$ CFD evaluations.

*Data availability.* Data is available upon request to corresponding author.

*Acknowledgements.* We would like to thank the members at the MDO Lab for interest and support, in particular Eirikur Jonsson for assistance with programs in the MACH framework, Nicholas Bons for helping with pyGeo and Anil Yildirim for his assistance with the ANK solver. We also thank research associate Charles Mader for consulting on the AD improvements implemented in the adjoint solver. Finally, researcher Michael McWilliam at DTU kindly assisted with IEA material.



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
