# Peer review of "Multipoint high-fidelity CFD-based aerodynamic shape optimization of a 10 MW wind turbine"

_Wind Energy Science, 2018_

## Referee Comment (RC1) · Anonymous Referee #1 · 22 Nov 2018

The presented paper gives an extensive overview on the current state of the art in optimization based on CFD. Afterwards, own research on this topic is presented and an interesting overview on the optimization of rotor blades using single-point and multi-point optimization is presented. The results and used tool chain are very interesting and worth publishing.

The main concerns when reading this manuscript are regarding the chosen structure, some presented topics that are out of the scope of this paper and there are some unclear conclusions. Therefore, I suggest a major revision before publication.

I highly recommend omitting the part comparing two CFD codes, as this is not within the scope of the paper. Instead, some further information on the implementation and usage of the adjoints are highly recommended.

The review is structured as follows: First, some general remarks on the chosen structure and order of chapters and sections are given. Second, the content is discussed and finally typos, wrong placing and further minor errors are listed.

Note: Sometimes, there were inconsistent line numbers (starting with 20 at the page top for example). In these cases, I still used the line numbering from the manuscript for my comments.

Structure:

1.1 – 1.2.3 cover related literature, but only 1.1 is titled related literature. The structure should be corrected. 1.2.1 is titled airfoil optimization and 1.2.2 blade optimization. But 1.2.3 also shows airfoil and blade optimization. Another structure can help the reader to properly differ between the topics.

Page 6 Line 4-15: This paragraph is confusing. I suggest to rather sort it by topic then by papers above and below. Lines 10-13 therefore are unclear: What exactly is effective, which computational effort can be reduced under which circumstance – based on which publication.

1.3 and 1.4 could be combined. First give an overview of the present work, then add details on the chosen case.

For the sake of clarity, a consistent notation of the convergence limit ($10*10^{-5}$ vs 1.0e-4) should be used within the text and plots throughout the paper.

Section 5.2: You introduce the comparison to BEM, then present the results of high-fidelity optimization and later the results of the comparison to BEM. This should be reordered.

Page 33 L 7-14: Should be resorted. Jumps between problem, solution and findings are made.

Content:
[Figure]

Page 1 Line 7-8: The authors claim that especially the tip and root region can gain from CFD based optimization, as BEM is not accurate enough in these regions. This is again stated in Line 14. Nevertheless, the usage of gradient based methods, which is the only used method in this work, also requires very good convergence of the flow field. It is known that these regions are difficult for RANS solvers, as instationary effects, separation and vortices are present. So, it is at least to mention, that also this set-up comes with difficulties.

Page 8 Line 11: The statement that your cases converge typically well below 1.0e-4 is not true. Figures 19 and 22 show other behavior. Please differentiate more on that.

Page 10 Overview: The discussion of the optimizations and comparison misses the information about the flow. The used velocities / Reynolds numbers are highly important for convergence of gradients. This should be mentioned when comparing mesh sizes and design cycles.

Page 11 Line 26-28: It should get cleared at this point, if an initial movement of the mesh is done based on the chosen parametrization or not.

Page 13 Line 21-22: The favorable comparison you cite between EllipSys3D and Open-FOAM deals with atmospheric flow over complex terrain. It is not possible to judge based on this paper how the two solvers behave when looking at aerodynamics – which is the topic in this paper. The final report of IEA Wind Task 29 Phase 3 shows that both solvers lead to comparable results in aerodynamics. Therefore this statement should be adjusted.

Page 14 Section 2.3: This section lacks information. You don't cite any other work with ADjoint – it seems to be the first published work with adjoint gradients. So, some detail on the implementation and verification of gradients should be added.

Chapter 3: An extensive revision of this chapter is suggested. It should focus on the numerical Set-Up and chosen WEA. A comparison of two CFD codes (plus one BEM

code) is out of the scope of this paper and should be taken out. The validation could be topic of another publication, which would also allow to have a more detailed look on blade parts and flow phenomena, where incompressible and compressible solver work differently and lead to different results.

In section 3.1 "Computational Mesh" you talk about the WEA model you use, while the mesh is merely topic in this section. You have to justify, based on the results from the GCI, why you keep using mesh L2 (especially when looking at figure 5). The use of hotstarts in your work should be explained here (it was mentioned on page 5 that you will use them, but here it should be explained how). Otherwise it is hard to understand why you should keep using L2.

Page 23 L15-16: It is not clear, which criteria you used to choose the box resolution. Only the two at the blade root are justified, the following testing and choice is not explained at all. Please elaborate more on the choices.

Page 24 L 7-8: It is not clear which parameter sweeps where conducted. Which parameters where swept in which ranges?

Page 26 Table: Another line should be added containing the total CPU time and Iterations for optimizations using a hotstart.

Page 31 Figure caption: You mention hotstarts for L0 and L1. Up to here, only hotstarts for L0 where used. This should be handled consistently throughout the manuscript.

Page 30 L 17-19: The statement of "more or less doubled" is exaggerated as factors between 1.42 and 1.96 are shown in table 8. And the largest improvement is found for the coarse mesh, which should not be the reference for the assessment. This again comes up in the conclusion on page 36, Line 10.

Page 33 5.3.1: The finding should be discussed. What movements are suppressed by this constraint? What makes the results more realistic?

Page 34 Figure 22: All optimizations seem to stop with a higher value for the optimality

than some steps before. What is the stopping criterium for cases where convergence of the optimization is not reached? The same can be seen in Fig. 19 (left) for L1. Please explain this behavior.

Further:

Generally: Along the whole document the usage of "x%" is not consistent, also units are used in various ways. Examples are p.9 L 5, p. 15 table caption, p. 26 m/s and m are used in various notations.

Page 1 Line 1-2: Sentence is unclear

Page 2 Line 26-28: Citation is bad placed.

Page 2 Line 28: Start of sentence wrong

Page 5 Line 24: ". . .. make use of a surrogate. . ."

Page 10 Caption: ".... variety of algorithms against we refrain. . ."

Page 12 Caption: ". . . iterative operations that take up. . ."

Page 13 L 8: ". . . (all stored at cell centers). . .."

Page 17 L 15: Empty line

Page 18 L 27: ". . . mesh level. It is also. . ."

Page 19 L 2: ". . .results can be seen save perhaps for the. . ."

Page 25 L 10 and caption Fig. 12: Wrong reference

Page 26 L 22-24: Somewhere the name BEM2 should be introduced. Maybe "The second BEM optimization BEM2 covers the wind speeds . . .."

Page 27 L 8: "Fig. 16 shows the convergence history. . ."

Page 27 Figure 16: Third line should be named

Page 27 Figure 17 caption: Last sentence should be reordered

Page 28 Figure 18 caption: Number and unit are in separated lines

Page 29 Line 2 – 5: Sentence not clear.

Page 29 Line 7: "... in final designs obtained..."

Page 29 Empty line

Page 29 Equation 10: P = T · $\omega$ should be given, otherwise "Using values for torque from Tab. 7" is not connected to the equation.

Page 30 Empty line

Page 35 Figure 24: Might be an optical illusion, but the plots rather look like a comparison between Singlepoint and Multipoint with LE Constraint instead of Multipoint with and without LE constraint. Please check again.

---

## Referee Comment (RC2) · Anonymous Referee #2 · 3 Dec 2018

This paper presents an adoint-based method optimisation of a wind turbine blade under a number of optimisation conditions. Convergence is shown to occur for a range of mesh resolutions, and within the constraints optimum designs are found. The article focuses on the comparison between low order (BEM) with high order (CFD) solutions. More focus on medium-order solutions would improve the general literature review of the article. Both single point and multi-point optimisation is carried out. The multipoint optimisation of the surface is carried out, which outlines the relevance and importance of the contribution, as it demonstrates a tool which completely optimises the aerodynamic shape.

General comments:

- Lines 10-20, restatement multiple times of advantages of CFD over BEM - Section

1.1: No mention is made of optimizations using medium-order fidelity tools. Is this deliberate or was little material found? - Computational resources and times would be appreciated to allow a connection to designing engineers.

Specific comments:

- Line 24: In the reviewer's experience, the role of a winglet is exactly that: to reduce induced drag on a lifting body. This hence is not necessarily surprising. - Page 16: Line 12: More specifically: Given that the analysis here is carried out on a rigid geometry, tower influence can likely be neglected. - The reviewer believes the discrepancy in the optimum twist angle is likely due to the result of making use of specified polar data for the BEM1 optimisation (without having read the reference...) - Please provide more details on the multipoint optimisation, particularly the profile optimisation parameters.

Technical corrections:

- English: naive (multiple locations) - Page 4: Ln 21: "an 11% increase..." - Page 25: Line 10 Table ?? - Page 25: Figure 12: Tab ?? - Page 35: Line 14: superfluous

WESD

---

## Editor Comment (EC1) · Bianchini (Editor) · 14 Dec 2018

Dear authors, the Interactive Discussion phase is now over. You are encouraged to answer to Reviewers' comments carefully at your earliest convenience. Best regards

---

## Author Comment (AC1) · 29 Jan 2019

**Multipoint high-fidelity CFD-based aerodynamic shape optimization of a 10 MW wind turbine *Wind Energy Science* Manuscript ID: https://doi.org/10.5194/wes-2018-66**

Mads H. Aa. Madsen and Frederik Zahle and Niels N. Sørensen and Joaquim R. R. A. Martins

NOTE: Actions taken are identified in this color.

**Reviewer 1 Comments and Response**

**Comment 1:** "The presented paper gives an extensive overview on the current state of the art in optimization based on CFD. Afterwards, own research on this topic is presented and an interesting overview on the optimization of rotor blades using single-point and multi-point optimization is presented. The results and used tool chain are very interesting and worth publishing."

**Response:** Thank you for the review and interest in both results and tools. We find the review particularly thorough and are thankful for the reviewer's suggestions.

**Comment 2:** "The main concerns when reading this manuscript are regarding the chosen structure, some presented topics that are out of the scope of this paper and there are some unclear conclusions. Therefore, I suggest a major revision before publication."

**Response:** We agree, that the structure proposed by the reviewer improves the paper. We adopted the structure suggested by the reviewer (more detailed response below). The only point we do not fully agree with is that the flow solver comparison is outside the scope of the present paper. However, we took this opinion in consideration. We have moved the entire validation to the appendix except for the grid study, which is crucial when considering which grid level to use, and reduced this material. We do agree that some conclusions were unclear and followed the reviewer's the suggestions. The rephrased some conclusions for better clarity.

**Comment 3:** "I highly recommend omitting the part comparing two CFD codes, as this is not within the scope of the paper. Instead, some further information on the implementation and usage of the adjoints are highly recommended."

**Response:** We concur with the reviewer that the implementation of the adjoint solver is a crucial topic and that the literature for adjoint solvers dedicated to wind energy applications is sparse. The details of the adjoint solver implementation are too involved and out of the scope of the current paper. This is a separate contribution that does not involve the authors of the present paper. A separate paper is currently being prepared on the adjoint implementation by another set of authors that includes the last author of the present paper. The scope of the present paper is the state-of-the-art *application* of the adjoint methodology

on a modern, industrially relevant turbine (which we have not found elsewhere). We elaborated on the implementation as desired by the reviewer (highlighted in yellow in Section 3.2.2)

**Comment 4:** "The review is structured as follows: First, some general remarks on the chosen structure and order of chapters and sections are given. Second, the content is discussed and finally typos, wrong placing and further minor errors are listed."

**Response:** Thank you for the clear and thorough suggestions. Each item has been addressed with our comments in blue and actions taken in orange.

**Comment 5:** Structure comment: "1.1-1.2.3 cover related literature, but only 1.1 is titled related literature. The structure should be corrected. 1.2.1 is titled airfoil optimization and 1.2.2 blade optimization. But 1.2.3 also shows airfoil and blade optimization. Another structure can help the reader to properly differ between the topics."

**Response:** We agree with the reviewer regarding the titles in the various section of the review. The section titles have been revised. We also improved the comments on a number of works (major changes highlighted in yellow) and added small paragraph before Section 2.1 to explain the structure of the review. As is now evident from the section titles, it is *deliberate* that, for example airfoil optimization is found both in Section 2.2 and in Section 2.3, since airfoils have been optimized both with and without using the adjoint method. The distinction was made so that the reader can get a compressed overview of CFD adjoint-based aerodynamic optimization for wind energy applications in Section 2.3.

**Comment 6:** Structure comment: "Page 6 Line 4-15: This paragraph is confusing. I suggest to rather sort it by topic then by papers above and below. Lines 10-13 therefore are unclear: What exactly is effective, which computational effort can be reduced under which circumstancebased on which publication."

**Response:** L. 4-15 We re-wrote the entire paragraph (highlighted in yellow). L. 10-13 The lines covering, what is effective, computational effort and related circumstances have been removed as they muddied the overall message. The resulting, revised text is the last two paragraphs before Section 2.3 starting from "To optimize with respect to ...".

**Comment 7:** Structure comment: "1.3 and 1.4 could be combined. First give an overview of the present work, then add details on the chosen case."

**Response:** The content from the previous Section 1.3 has been removed completely. It was found to be superfluous. The content from the previous Section 1.4 has been moved. It is now just before Section 2.

**Comment 8:** Structure comment: "For the sake of clarity, a consistent notation of the convergence limit $(10 * 10^5$ vs $1.0e - 4)$ should be used within the text and plots throughout the paper."

**Response:** The reviewer is correct. Done as requested using the format: $10 \cdot 10^{-5}$.

**Comment 9:** Structure comment: "Section 5.2: You introduce the comparison to BEM, then present the results of high-fidelity optimization and later the results of the comparison to BEM. This should be reordered."

**Response:** Section 6.2 on planform optimization was reordered and is now clearly divided: First, the high-fidelity results for the planform optimization are presented for all mesh levels. Then, a comparison with the BEM results is given at the end of Section 6.2 starting from "We now compare our " on p. 25.

**Comment 10:** Structure comment: "Page 33 L 7-14: Should be resorted. Jumps between problem, solution and findings are made."

**Response:** The line numbering on the mentioned page was faulty. We assume that the reviewer is referring to the text section just before Section 5.3.1 (old manuscript) on the performance of the optimized design. The critiqued paragraph was rephrased and reduced (highlighted in yellow). See the last two paragraphs of Section 6.3 starting from "Comparing the airfoil shapes" on p. 28.

**Comment 11:** Content comment: "Page 1 Line 7-8: The authors claim that especially the tip and root region can gain from CFD based optimization, as BEM is not accurate enough in these regions. This is again stated in Line 14. Nevertheless, the usage of gradient based methods, which is the only used method in this work, also requires very good convergence of the flow field. It is known that these regions are difficult for RANS solvers, as instationary effects, separation and vortices are present. So, it is at least to mention, that also this set-up comes with difficulties."

**Response:** Due to a comment by another reviewer the abstract has changed. We removed the redundant mentioning of tip and root as critical areas in the abstract. The reviewer is correct that gradient based methods require good convergence of the flow field. This was also pointed out on p. 6 (in the original manuscript). We emphasize the point made by the reviewer on p. 13 in Section 3.2.2 (highlighted in yellow).

**Comment 12:** Content comment: "Page 8 Line 11: The statement that your cases converge typically well below 1.0e-4 is not true. Figures 19 and 22 show other behavior. Please differentiate more on that."

**Response:** We changed the lines concerning the convergence of the design optimization problem. See last line of penultimate paragraph on p. 9. Table 2 was also changed. We did not give illustrations of cases with deeper convergence as seen in Fig. 1, as it was found superfluous.

[Figure]

Figure 1: Example of deep convergence

**Comment 13:** Content comment: "Page 10 Overview: The discussion of the optimizations and comparison misses the information about the flow. The used velocities / Reynolds numbers are highly important for convergence of gradients. This should be mentioned when comparing mesh sizes and design cycles."

**Response:** The point the reviewer makes about convergence being important for gradients was already in the original manuscript on p. 6, l 4-5. We added a new column to include Reynolds numbers in Table 1.

**Comment 14:** Content comment: "Page 11 Line 26-28: It should get cleared at this point, if an initial movement of the mesh is done based on the chosen parametrization or not."

**Response:** There is no initial movement of the mesh since no gradient has been computed yet. A clarification has been inserted (highlighted in yellow). See start of Section 3.

**Comment 15:** Content comment: "Page 13 Line 21-22: The favorable comparison you cite between EllipSys3D and OpenFOAM deals with atmospheric flow over complex terrain. It is not possible to judge based on this paper how the two solvers behave when looking at aerodynamics which is the topic in this paper. The final report of IEA Wind Task 29 Phase 3 shows that both solvers lead to comparable results in aerodynamics. Therefore this statement should be adjusted."

**Response:** The reviewer is correct. Indeed, the comparison by Cavar et al. does also state that OpenFOAM and EllipSys can produce "almost identical" numerical results. The solver comparison does however state, that a speed difference was observed "in achieving numerical results of the same order of accuracy". The word 'favourable' was removed. It was clarified that the results from the two solvers were "almost identical" (quote from the comparison by Cavar et al.). The IEA Wind Task 29 Phase 3 (and two other references) were added as references to substantiate the claim. It is mentioned that a speed difference was observed in the comparison by Cavar et al. All changes are highlighted in yellow at the end of Section 3.2.1.

**Comment 16:** Content comment: "Page 14 Section 2.3: This section lacks information. You don't cite any other work with ADjoint - it seems to be the first published work with adjoint gradients. So, some detail on the implementation and verification of gradients should be added."

**Response:** This paper was not intended to present a detailed description of the adjoint implementation because this was a contribution by another set of authors (which includes the last author or the present paper.) Nevertheless, we made some changes in Section 3.2.2 (highlighted in yellow). We cite the previous work, which has more details. Another paper is currently being prepared with more details on the adjoint implementation. To better reflect the fact that we improved an already existing adjoint solver that has been used in ADflow for several years, we made the following changes: The content of Section 2.3 'ADjoint: A discrete adjoint solver architecture' was added to the end of Section 3.2.2 'ADflow'. Also the outer section 2.2 'Flow solvers' was renamed to: 3.2 'Flow and adjoint solvers'. The implementation has been further explained (highlighted in yellow in Section 3.2.2). However, we do not plan to also include further gradient verification as that has been the subject of other publications [3].

**Comment 17:** Content comment: "Chapter 3: An extensive revision of this chapter is suggested. It should focus on the numerical Set-Up and chosen WEA. A comparison of two CFD codes (plus one BEM code) is out of the scope of this paper and should be taken out. The validation could be topic of another publication, which would also allow to have a more detailed look on blade parts and flow phenomena, where incompressible and compressible solver work differently and lead to different results."

**Response:** The CFD solver comparison has been moved to appendix A with the exception of the grid convergence study, which is crucial in order to discuss which grid level to select for the results section.

**Comment 18:** Content comment: "In section 3.1 Computational Mesh you talk about the WEA model

you use, while the mesh is merely topic in this section. You have to justify, based on the results from the GCI, why you keep using mesh L2 (especially when looking at figure 5). The use of hotstarts in your work should be explained here (it was mentioned on page 5 that you will use them, but here it should be explained how). Otherwise it is hard to understand why you should keep using L2."

**Response:** We renamed 'Section 3.1 Computational Mesh' to 'Section 4.1 Fluid model and computational mesh' to reflect the point made by the reviewer. It is easy for us to justify the use of mesh level L2: It is one of the interesting findings in the paper that such a mesh level can lead to even *wrong* design trends and thus should be avoided or at least used carefully. In order to justify this point for interested readers we of course have to show how the mesh performs. We inserted a paragraph in 'Sec. 4.2 Mesh convergence study' to explain why we (also) show results from L2 and why L2 should be used carefully. Note: the term 'hotstart' has been substituted with 'warm start' since it seems prevalent in the SNOPT manual. We elaborated on the use of warm starts as requested by the reviewer. The elaboration was moved to Section 6.1 Pitch optimization (highlighted in yellow).

**Comment 19:** Content comment: "Page 23 L15-16: It is not clear, which criteria you used to choose the box resolution. Only the two at the blade root are justified, the following testing and choice is not explained at all. Please elaborate more on the choices."

**Response:** We added a discussion on the box resolution in Section 5.2 as requested (highlighted in yellow).

**Comment 20:** Content comment: "Page 24 L 7-8: It is not clear which parameter sweeps where conducted. Which parameters where swept in which ranges?"

**Response:** A clarification has been inserted in Section 6.1 (highlighted in yellow).

**Comment 21:** Content comment: "Page 26 Table: Another line should be added containing the total CPU time and Iterations for optimizations using a hotstart."

**Response:** A clarification has been inserted in the footnotes of Table 6 (highlighted in yellow).

**Comment 22:** Content comment: "Page 31 Figure caption: You mention hotstarts for L0 and L1. Up to here, only hotstarts for L0 where used. This should be handled consistently throughout the manuscript."

**Response:** In the original manuscript there is a hotstart (now called warm start) mentioned from L2 to L1 on p. 27, fig. 17. We inserted a clarifying paragraph (highlighted in yellow). It is the penultimate paragraph of Section 6.1.

**Comment 23:** Content comment: "Page 30 L 17-19: The statement of more or less doubled is exaggerated as factors between 1.42 and 1.96 are shown in table 8. And the largest improvement is found for the coarse mesh, which should not be the reference for the assessment. This again comes up in the conclusion on page 36, Line 10."

**Response:** The critiqued statement was removed. The actual improvement factor for the L0 mesh was inserted in the conclusion (highlighted in yellow).

**Comment 24:** Content comment: "Page 33 5.3.1: The finding should be discussed. What movements are suppressed by this constraint? What makes the results more realistic?"

**Response:** We added more discussion on this finding addressing the reviewer's points (highlighted in yellow). See Section 6.4.

**Comment 25:** Content comment: "Page 34 Figure 22: All optimizations seem to stop with a higher value for the optimality than some steps before. What is the stopping criterium for cases where convergence of the optimization is not reached? The same can be seen in Fig. 19 (left) for L1. Please explain this behavior."

**Response:** The behavior of SNOPT to stop at a (slightly) higher optimality than previous values can occur - but only when the convergence is not met. This behavior is well known and can be found in the literature (take e.g. [1, Fig. 6]). SNOPT has many possible exit codes for a premature stop when convergence is not reached. We typically see exit code 40 (`terminated after numerical difficulties`) and info code 41 (`current point cannot be improved`). Usually, when SNOPT cannot reach a desired threshold it exits with exit code 40 (`terminated after numerical difficulties`) and info code 41 (`current point cannot be improved`). When this happens, SNOPT can choose to stop on a value higher than the minimum found value. For further details on SNOPT [2] please see the manual [1]. We inserted a discussion of this behavior (Sec. 6.3 in revised manuscript).

**Comment 26:** "Generally: Along the whole document the usage of x% is not consistent, also units are used in various ways. Examples are p.9 L 5, p. 15 table caption, p. 26 m/s and m are used in various notations."

**Response:** All points have now been corrected as specified below giving a fully consistent notation throughout the document. All percentages have been put into the chosen format. All units have been put into the chosen format. P. 26 is also fixed.

**Comment 27:** "Page 1 Line 1-2: Sentence is unclear"

**Response:** Fixed.

**Comment 28:** "Page 2 Line 26-28: Citation is bad placed."

**Response:** Fixed. The relevant line is now on p. 2.

**Comment 29:** "Page 2 Line 28: Start of sentence wrong"

**Response:** The two sentences were rephrased. See p. 2.

**Comment 30:** "Page 5 Line 24: . . .. make use of a surrogate. . ."

**Response:** This seems to be Page 5 Line 19 in the original pdf. We rephrased the sentence.

**Comment 31:** "Page 10 Caption: .... variety of algorithms against we refrain. . ."
* * *
[1] http://www.sbsi-sol-optimize.com/manuals/SNOPT%20Manual.pdf

**Response:** Fixed.

**Comment 32:** "Page 12 Caption: ... iterative operations that take up..."

**Response:** Fixed.

**Comment 33:** "Page 13 L 8: ...(all stored at cell centers)..."

**Response:** Fixed.

**Comment 34:** "Page 17 L 15: Empty line"

**Response:** Fixed.

**Comment 35:** "Page 18 L 27: ...mesh level. It is also..."

**Response:** Fixed.

**Comment 36:** "Page 19 L 2: . . .results can be seen save perhaps for the. . ."

**Response:** Fixed.

**Comment 37:** "Page 25 L 10 and caption Fig. 12: Wrong reference"

**Response:** Fixed.

**Comment 38:** "Page 26 L 22-24: Somewhere the name BEM2 should be introduced. Maybe The second BEM optimization BEM2 covers the wind speeds..."

**Response:** The sentence has been clarified as requested. Furthermore, Section 5.1 now clearly defines all CFD and BEM design optimization problems.

**Comment 39:** "Page 27 L 8: Fig. 16 shows the convergence history..."

**Response:** Fixed.

**Comment 40:** "Page 27 Figure 16: Third line should be named"

**Response:** The figure has been updated as requested (see Fig. 11).

**Comment 41:** "Page 27 Figure 17 caption: Last sentence should be reordered"

**Response:** Agreed. Rephrased.

**Comment 42:** "Page 28 Figure 18 caption: Number and unit are in separated lines"

**Response:** Fixed.

**Comment 43:** "Page 29 Line 2-5: Sentence not clear."

**Response:** Agreed. The sentences have been rephrased.

**Comment 44:** "Page 29 Line 7: . . . in final designs obtained. . ."

**Response:** Fixed.

**Comment 45:** "Page 29 Empty line"

**Response:** Fixed.

**Comment 46:** "Page 29 Equation 10: $P = T \cdot w$ should be given, otherwise Using values for torque from Tab. 7 is not connected to the equation."

**Response:** Agreed. The relevant equation is now eq. 9. The relation between power and torque was added to Section 5.1 where a description of power is needed for the first time.

**Comment 47:** "Page 30 Empty line"

**Response:** Fixed.

**Comment 48:** "Page 35 Figure 24: Might be an optical illusion, but the plots rather look like a comparison between Singlepoint and Multipoint with LE Constraint instead of Multipoint with and without LE constraint. Please check again."

**Response:** The reviewer is correct. The figure was updated to show data from multipoint simulations with and without LE constraint as requested.

**Reviewer 2 Comments and Response**

**Comment 1:** "This paper presents an adjoint-based method optimisation of a wind turbine blade under a number of optimisation conditions. Convergence is shown to occur for a range of mesh resolutions, and within the constraints optimum designs are found. The article focuses on the comparison between low order (BEM) with high order (CFD) solutions. More focus on medium-order solutions would improve the general literature review of the article. Both single point and multi-point optimisation is carried out. The multipoint optimisation of the surface is carried out, which outlines the relevance and importance of the contribution, as it demonstrates a tool which completely optimises the aerodynamic shape."

**Response:** Thank you for the review and the compliment. Indeed, medium-order solutions were hardly covered. As noted below, we have added references on medium-fidelity for the review to be more general as suggested by the reviewer. See p. 4.

**Comment 2:** General comment: "- Lines 10-20, restatement multiple times of advantages of CFD over BEM "

**Response:** We removed the redundant mentioning of tip and root as critical areas in the abstract.

**Comment 3:** General comment: "- Section 1.1: No mention is made of optimizations using medium-order fidelity tools. Is this deliberate or was little material found?"

**Response:** We improved the literature review by including several references (highlighted in yellow in Section 2.1).

**Comment 4:** General comment: "- Computational resources and times would be appreciated to allow a connection to designing engineers."

**Response:** Benchmarking codes and timings are certainly crucial topics. We do not focus on it in this paper, but we do mention the CPU times for the pitch optimizations in Table 6 for interested readers, were we can see that the L2 optimization took 106.9 CPU hours. This means that we with 216 procs could complete the optimization in: 106.9 h / 216 = 0.5 h, i.e. 30 minutes. We added an explanation of the CPU timing in the table caption (highlighted in yellow).

**Comment 5:** Specific comment: "- Line 24: In the reviewers experience, the role of a winglet is exactly that: to reduce induced drag on a lifting body. This hence is not necessarily surprising."

**Response:** We agree with the reviewer that the notion was redundant. Sentence was removed.

**Comment 6:** Specific comment: "- Page 16: Line 12: More specifically: Given that the analysis here is carried out on a rigid geometry, tower influence can likely be neglected."

**Response:** Correct. We added the point made by the reviewer (highlighted in yellow). See second sentence in Section 4.1.

**Comment 7:** Specific comment: "- The reviewer believes the discrepancy in the optimum twist angle is likely due to the result of making use of specified polar data for the BEM1 optimisation (without having read the reference...)"

**Response:** Both BEM1 and BEM2 use specified polar data. The difference is that BEM2 also can change the relative thickness by interpolating between the available airfoils. Clarification inserted in manuscript. See end of Section 5.1.

**Comment 8:** Specific comment: "- Please provide more details on the multipoint optimisation, particularly the profile optimisation parameters."

**Response:** We altered Section 5.1 to clearly define the single point and multipoint optimizations. Further motivation was added to the multipoint section (Sec. 6.4). We also added a visualisation (Fig. 16) of why the chosen wind speeds incur more than one AOA.

**Comment 9:** Technical comment: "- English: naive (multiple locations)"

**Response:** Fixed.

**Comment 10:** Technical comment: "- Page 4: Ln 21: an 11% increase. . ."

**Response:** Fixed.

**Comment 11:** Technical comment: "- Page 25: Line 10 Table ??"

**Response:** Fixed.

**Comment 12:** Technical comment: "- Page 25: Figure 12: Tab ??"

**Response:** Fixed.

**Comment 13:** Technical comment: "- Page 35: Line 14: superfluous"

**Response:** Fixed.

**Other modifications**

- The notation of units was inconsistent. This has been fixed.
- We added a recent reference on the ANK solver for interested readers.
- We added Table 3 so that the reader easily can compare the cited literature.
- Table 5 was improved.
- Figure 20 now includes multipoint results.
- Table 9 was added to allow the reader to easily get an overview of the main results.

**References**

[1] Chen, Z Lyu, GKW Kenway, and JRRA Martins. Aerodynamic shape optimization of the Common Research Model wing-body-tail configuration. *Journal of Aircraft*, 53:276–293, January 2016.

[2] PE Gill, W Murray, and MA Saunders. Snopt: An sqp algorithm for large-scale constrained optimization. *Siam Journal on Optimization*, 12(4):979–1006, 2002.

[3] CA Mader. ADjoint: An approach for the rapid development of discrete adjoint solvers. 2007.